# The posterior parietal cortex contributes to visuomotor processing for saccades in blindsight macaques

Rikako Kato [1,2], Takuya Hayashi [3,4], Kayo Onoe[3,4], Masatoshi Yoshida[1,5,10], Hideo Tsukada[6], Hirotaka Onoe[4,7], Tadashi Isa[1,2,5,7,8 ✉] & Takuro Ikeda [1,9 ✉]

Patients with damage to the primary visual cortex (V1) lose visual awareness, yet retain the ability to perform visuomotor tasks, which is called "blindsight." To understand the neural mechanisms underlying this residual visuomotor function, we studied a non-human primate model of blindsight with a unilateral lesion of V1 using various oculomotor tasks. Functional brain imaging by positron emission tomography showed a significant change after V1 lesion in saccade-related visuomotor activity in the intraparietal sulcus area in the ipsi- and contralesional posterior parietal cortex. Single unit recordings in the lateral bank of the intraparietal sulcus (lbIPS) showed visual responses to targets in the contralateral visual field on both hemispheres. Injection of muscimol into the ipsi- or contralesional lbIPSs significantly impaired saccades to targets in the V1 lesion-affected visual field, differently from previous reports in intact animals. These results indicate that the bilateral lbIPSs contribute to visuomotor function in blindsight.

[1] Department of Developmental Physiology, National Institute for Physiological Sciences, Okazaki, Japan. [2] Department of Neuroscience, Graduate School of Medicine, Kyoto University, Kyoto, Japan. [3] Laboratory for Brain Connectomics Imaging, RIKEN Center for Biosystems Dynamics Research, Kobe, Japan. [4] Center of Molecular Imaging Science, RIKEN, Kobe, Japan. [5] School of Life Sciences, the Graduate University of Advanced Studies (SOKENDAI), Hayama, Japan. [6] Central Research Laboratory, Hamamatsu Photonics K.K., Hamamatsu, Japan. [7] Human Brain Research Center, Graduate School of Medicine, Kyoto University, Kyoto, Japan. [8] Institute for the Advanced Study of Human Biology (WPI-ASHBi), Kyoto University, Kyoto, Japan. [9] Department of Cognitive Neuroscience, Primate Research Institute, Kyoto University, Inuyama, Japan. [10]Present address: Center for Human Nature, Artificial Intelligence, and Neuroscience (CHAIN), Hokkaido University, Sapporo, Japan. ✉email: isa.tadashi.7u@kyoto-u.ac.jp; taikeda-ns@umin.ac.jp

The primary visual cortex (V1) is a gateway for cortical visual processing. Patients with damage in V1 lose their visual awareness in the corresponding visual field, however, they are often able to localize a visual target without visual awareness when asked to[1–3]. Such residual visuomotor ability is called "blindsight" and has attracted interest from researchers in various fields. Several lines of investigation have been conducted to explore the neural mechanisms of blindsight in nonhuman primates. Early studies characterized the residual visuomotor capacities, such as pupil responses[4], visually guided saccades[5], and forearm reaching[6]. Furthermore, Cowey and Stoerig demonstrated that these monkeys lose visual awareness in a similar way to human patients[7]. Our laboratory has devoted years to understand the residual visuomotor functions of monkeys with a V1 lesion. We have revealed the following impairments in blindsight: decisions become less deliberate[8], some aspects of spatial attention are impaired[9], and visual awareness is lost[10]. On the other hand, we found that monkeys retain nonreflexive complex functions, such as short-term spatial memory[11], probability-based spatial attention[12], and reward-based learning[13]. These results suggest that blindsight is not just a simple low-level response, and the ability to perform various types of complex and cognitive behaviors is retained. Thus, it is reasonable to think that the residual visual pathways have access to the frontal/parietal cortices, which are responsible for the cognitive function. However, how this is possible and which region is responsible for the process remains unknown.

Extrastriate cortical areas, such as the middle temporal (MT)[14,15] and medial superior temporal (MST)[16] regions and the lateral intraparietal area (LIP) in the parietal cortex[17] show a clear visual response in the absence of V1. These visual responses are thought to be mediated by the superior colliculus (SC)[5,15,18] via the pulvinar and/or lateral geniculate nucleus[17,19]. However, these visual responses in the MT/MST/LIP were recorded under anesthesia or during a passive viewing state, and not during an active behavioral context. This leaves some questions, as the very nature of blindsight is the discrepancy between visual awareness and active localization[3]. Thus, it is necessary to study cortical activation, while subjects are participating in an active visuomotor task to understand fully the neural mechanisms underlying blindsight. For this purpose, we first performed whole-brain analysis in monkeys using positron emission tomography (PET) to visualize the areas, in which regional cerebral blood flow (rCBF) changed proportionally to the number of visuomotor events in a task. We then analyzed the differences in these visuomotor-related rCBF changes between the pre- and post-V1 lesion periods. Among other brain regions, the ipsi- and contralesional intraparietal sulcus (IPS) areas were found to be critically involved in visuomotor processing after the V1 lesion. Then, to test the causal function of these areas, we focused on the lateral bank of the intraparietal sulcus (lbIPS) which overlaps with LIP and conducted neurophysiological experiments. After confirming the visual responses of neurons in the lbIPSs, we injected muscimol, a GABA$_A$ receptor agonist, to either the ipsi- or contralesional lbIPSs. The results of the present study are very different from those of previous reports in intact monkeys, suggesting that the visuomotor function of bilateral lbIPSs is altered in blindsight.

## Results

**V1 lesion**. Primary visual cortices (V1) were unilaterally removed from three macaque monkeys (Fig. 1 and Supplementary Fig. 1). The monkeys showed clear deficits in making visually guided saccades to a target in their affected visual field immediately after lesioning. However, they regained their ability to localize a target in the affected field by saccades within a few months[8]. We

conducted the post-lesional experiments described below once their performance recovered to show residual visuomotor ability (>80% in the step saccade task at ~6 months after lesioning). Hereafter, we describe the brain areas ipsilateral and contralateral to the V1 lesion as "ipsilesional" and "contralesional," respectively. On the other hand, the visual field ipsilateral and contralateral to the V1 lesion is referred to as the "intact" and "affected" visual field, respectively, to avoid confusion.

**PET imaging**. We designed a behavioral task called the "round saccade task," in which the number of visuomotor events (i.e., visual target presentations and saccades) in a scanning session was controlled by task condition (R1, R2, R3, R4, or R6), while keeping the number of trials and reward amount constant. Two monkeys (monkey C: 13–15 trials/session; monkey T: 14–16 trials/session) were trained to perform the round saccade task (Fig. 2) and were examined in a PET scanner.

Before analyzing the PET results, we first compared saccade behavior between the pre- and post-lesion periods. Both monkeys could perform the task well even after V1 lesion (success rate in the pre-lesion period: 95% for monkey C and 96% for monkey T; success rate in the post-lesion period: 96% for monkey C and 95% for monkey T). We analyzed saccadic reaction times and endpoints (Supplementary Fig. 2). We could not find any consistent change specific to saccades to the affected visual field in the post-lesion period, except the variability of their endpoints. Saccade endpoints were distributed around the target in the pre- and post-lesion periods, but tended to be more broadly scattered in the affected visual field: the interquartile range was significantly larger than toward the intact field and in the pre-lesion period (Supplementary Fig. 2c). Note that although the timing and location of the target were predictable in the second and following rounds in each trial, the monkeys exhibited almost no express or anticipatory saccades (reaction time < 100 ms) in either the pre- or post-lesion period (<0.1% in both monkeys). The overall results showed that saccades to the affected visual field were qualitatively similar to those to the intact visual field, other than endpoint variability, which is consistent with our previous observation[8].

We then analyzed the rCBF data obtained by PET scanning. As the number of visuomotor events in a single scanning session was proportional to the task condition (R1, R2, R3, R4, and R6), we expected to see a significant relationship between rCBF and task condition if a region was involved in visuomotor processing (Tables 1 and 2). Thus, the differences in the task-related change in rCBF between the pre- and post-lesion periods will indicate the brain areas involved in altered visuomotor function in blindsight (Table 3). The areas with a significant relationship to task condition in the pre- and post-lesion periods are shown in Fig. 3a–d, respectively (see Supplementary Fig. 3 for individual analysis). The early visual cortices showed a significant relationship to task condition bilaterally in the pre-lesion period. This relationship completely disappeared in the ipsilesional hemisphere after lesioning, reflecting the physical loss of V1. In the post-lesion period, another cluster with a significant task relationship was found in the ipsilesional hemisphere that contained the IPS area, MT, and MST areas, which are related to visual/visuomotor function, suggesting that these areas are involved in residual visuomotor function in blindsight. Thus, to identify the regions involved in functional recovery after lesioning, we conducted whole-brain analysis to see the interaction effect between task and lesion (Fig. 4a, b). This showed a significant post-lesion increase of task relationship in the ipsilesional IPS area. In addition, a significant increase of task relationship was found in the contralesional IPS area. These

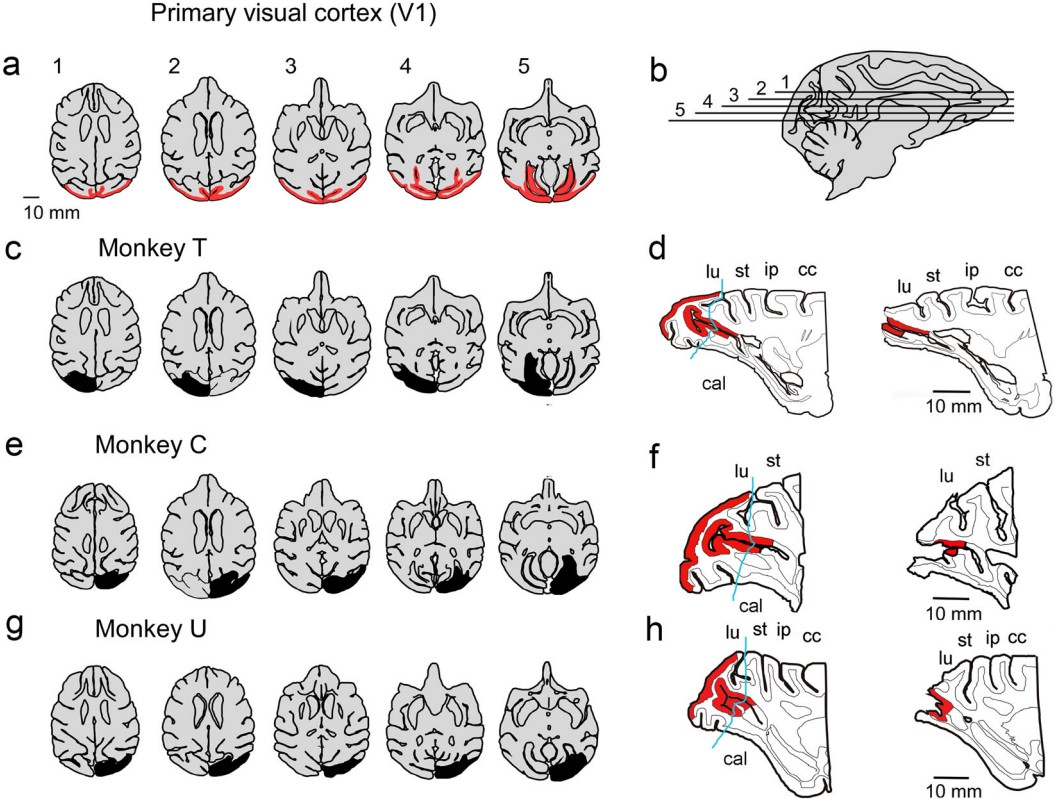

**Fig. 1 Lesion extent in the individual animals. a** The area of V1 in horizontal planes (colored in red). **b** Horizontal levels of the section in each panel of **a** with the corresponding numbers, overlaid on a sagittal brain slice. **c**, **e**, **g** The extent of the V1 lesion in monkeys T, C, and U is indicated on the horizontal planes in black, respectively. The horizontal levels of the individual horizontal sections in **c**, **e**, and **g** are matched with those in **a**. **d**, **f**, **h** The sagittal planes across the calcarine sulcus (cal) on the contralesional (left panel) and ipsilesional (right panel) side in monkey T, C, and U, respectively. The extent of V1 is indicated in red. Posterior (leftward) side of the blue lines (indicated in the arrow) in the left panels was considered to be lesioned on the ipsilesional side (right panel). cal calcarine sulcus, cc central sulcus, ip intraparietal sulcus, lu lunate sulcus, st superior temporal sulcus.

results suggest that the bilateral IPS areas have a greater role in visuomotor function in the post-lesion period than in the pre-lesion period. Regression of task relationship in ipsi- and contralesional IPS areas (Fig. 4c, d) supports the idea: regression coefficients were close to zero in pre-lesion period, but positive in post-lesion period in both ipsi- and contralesional IPS areas. These results suggest that the bilateral IPS areas were not critically involved in visuomotor processing in the pre-lesion period, but were involved in the post-lesion period. These findings are in line with the above results from whole-brain analysis and further imply that not only the ipsilesional, but the bilateral IPS areas contribute to the residual visuomotor function. However, because of the limited temporal resolution of the PET scanner, we could not distinguish between visual and motor function. For further understanding, we conducted physiological experiments targeting the bilateral IPS areas.

**Single-unit activity of lbIPS neurons after V1 lesion.** Among the IPS areas, it is known that its lateral bank (lbIPS) includes the regions related to the saccade control, such as LIP[20,21], therefore, we decided to focus on bilateral lbIPS. We sampled task-related neurons during an overlap saccade task to dissociate the visual and motor responses in ipsi- and contralesional lbIPSs in monkey U (Supplementary Fig. 4, blue rectangles). Seven and nine task-related neurons were recorded from the ipsi- and contralesional lbIPS, respectively. The small number of samples was due to instability of recordings in the animals with V1 lesion and also to our intention of minimizing injury to the area. The population activity of these neurons is illustrated in Fig. 5a, b. All seven

ipsilesional lbIPS neurons showed a clear phasic visual response when the target was presented in their response field (RF) in the affected field (Fig. 5a, red trace). The onset latency of the visual response was $141 \pm 21$ ms ($n = 7$, mean ± standard deviation [SD], Supplementary Fig. 5). In contrast, if the target was presented outside their RF in the intact visual field (blue trace), these neurons did not show a visual response. Similarly, all the nine contralesional lbIPS neurons showed a marked phasic visual response when the target was presented in their RF in the intact visual field (Fig. 5b, red trace). The mean onset latency of the visual response was $94 \pm 9$ ms ($n = 9$, mean ± SD), which was significantly shorter than that of the ipsilesional neurons ($p < 0.05$ by Welch $t$ test, Supplementary Fig. 5). Most of the lbIPS neurons showed sustained activity following the phasic visual response during the delay period (6/7 ipsi- and 8/9 contralesional neurons, $p < 0.05$ by $t$ test compared to the control period 1. See "Methods" section), although one ipsilesional lbIPS neuron showed reduced activity ($p < 0.05$ by $t$ test). Some neurons showed reduced activity during the delay period when the target was presented outside the RF (2/7 ipsi- and 3/9 contralesional neurons, $p < 0.05$ by $t$ test compared to control period 2. See "Methods" section).

Other than the difference in visual response latency, there was no clear difference between the ipsi- and contralesional lbIPS neurons (Supplementary Fig. 5). Ipsilesional lbIPS neurons tended to show slightly higher activity during the baseline period (control period 1) than contralesional lbIPS neurons (population average, ipsilesional lbIPS: $16.9 \pm 10.8$ spikes/s, contralesional lbIPS: $8.5 \pm 6.5$ spikes/s, $p = 0.07$ by $t$ test). There was no difference in the magnitude of the visual responses (difference

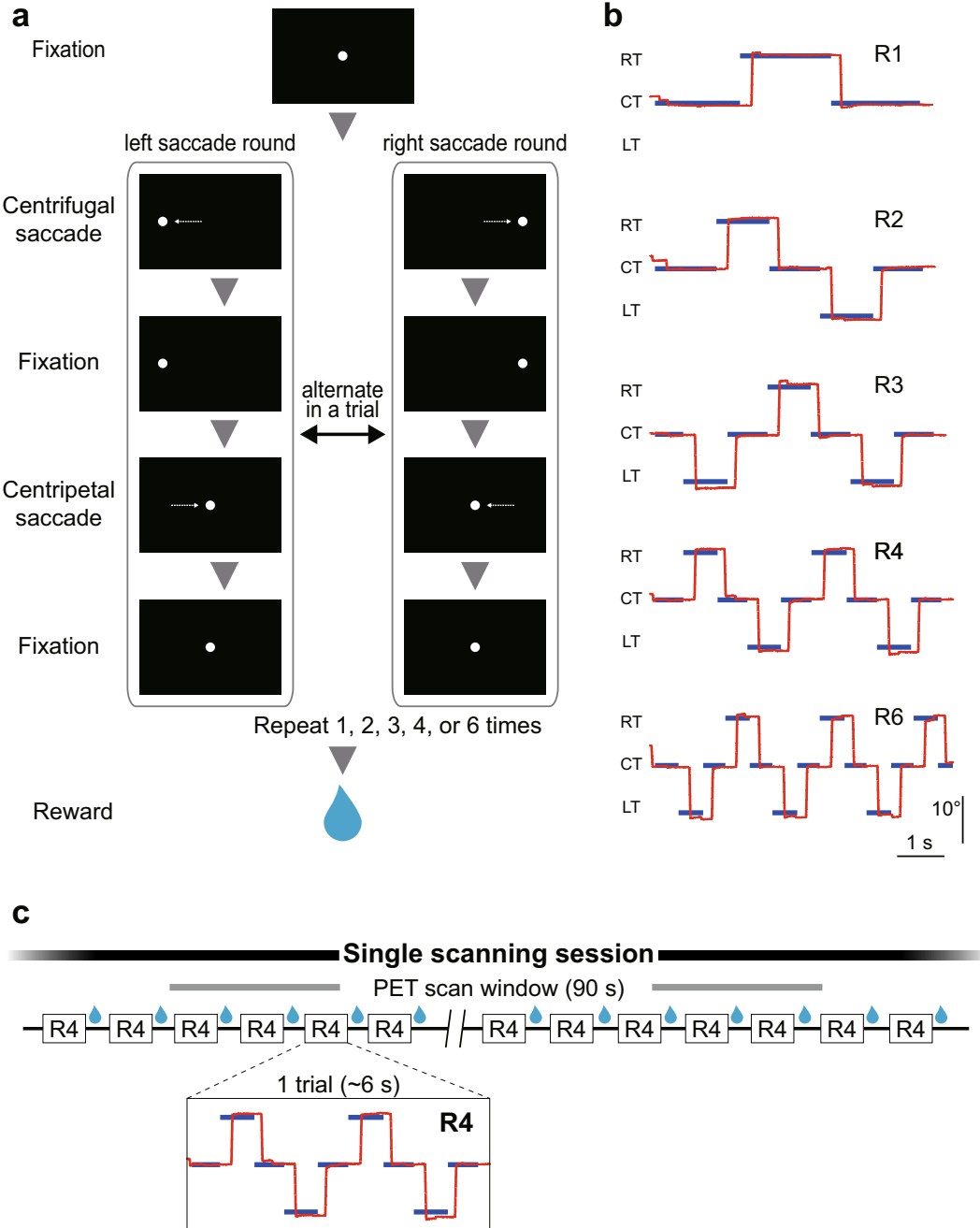

**Fig. 2 Task design: round saccade task. a** The round saccade task used in the PET imaging studies. A trial included 1, 2, 3, 4, or 6 rounds of saccades, determined in each task condition. **b** Example of trials in each condition. Vertical axis represents horizontal location on the monitor. Red lines indicate eye position, and blue lines indicate the locations of the target. Note that it took ~6 s to perform a single trial in all five task conditions. **c** A schematic drawing of a scanning session sequence. Each box indicates a single trial. Black lines indicate the intertrial intervals. A single scanning session (2 min) included ~20 trials regardless of task condition. CT central target, LT left target, RT right target.

between the peak visual responses and preceding baseline activity) between the ipsi- (22.1 ± 14.6 spikes/s) and contrale-sional lbIPS neurons (28.7 ± 11.4 spikes/s; $p = 0.33$ by $t$ test). The current results confirmed that ipsilesional lbIPS neurons retained these visual responses after V1 lesion, supporting the idea that the ipsilesional lbIPS is involved in residual visuomotor function in blindsight. Most lbIPS neurons showed higher activity around saccade onset when saccades were made to the target in their RF compared to the target outside their RF (5/7 ipsi- and 7/9 contralesional neurons, $p < 0.05$ by non-paired $t$ test), suggesting the existence of perisaccadic activity. The phasic presaccadic activity was not clear in Fig. 5. However, some neurons showed

presumed presaccadic activity in the step saccade task (see Supplementary Figs. 4 and 6).

**Effects of lbIPS inactivation after V1 lesion**. As described above, the lbIPS neurons exhibited clear visual responses during the overlap saccade task, followed by the sustained activity until saccade initiation, and we also found presaccadic activity in some neurons during step saccade task (Supplementary Fig. 4, blue squares and Supplementary Fig. 6). The existence of presaccadic activity and the MRI images of the electrode tracks estimated by the grid on the skull (see details in Supplementary Fig. 4) suggest

**Table 1 Whole-brain statistical analysis of rCBF changes related to task condition: pre-lesion period.**

**Pre-lesion**

$\beta > 0$

| Cluster index | Voxels | Z-max | COG (x, y, z) (mm) | Region | Local maxima (x, y, z) (mm) | Z-value | Locus |
|---|---|---|---|---|---|---|---|
| 1 | 91464 | 8.15 | −1.52, −16.1, 17.8 | Bilateral V1/V2/V3/V4 | −8.7, −19.7, 23.2 | 8.15 | Ipsilesional V1 |
| | | | | | −5.2, −19.2, 23.7 | 7.8 | Ipsilesional V2 |
| | | | | | 12.8, −16.7, 19.7 | 7.41 | Contralesional V1 |
| | | | | | 7.3, −21.2, 20.2 | 7.11 | Contralesional V1 |
| | | | | | 5.8, −19.7, 23.7 | 6.96 | Contralesional V1 |
| | | | | | −13.7, −9.74, 18.7 | 6.71 | Ipsilesional V1 |

$\beta < 0$

| Cluster index | Voxels | Z-max | COG (x, y, z) (mm) | Region | Local maxima (x, y, z) (mm) | Z-value | Locus |
|---|---|---|---|---|---|---|---|
| 1 | 12773 | 4.13 | 7.43, 22.8, 29.2 | Contralesional SMA/ACC/S1 | 1.8, 31.3, 24.2 | 4.13 | Contralesional ACC |
| | | | | | 20.8, 17.8, 25.2 | 3.98 | Contralesional S1 |
| | | | | | −0.2, 24.8, 33.2 | 3.83 | SMA |
| | | | | | 9.8, 14.3, 35.2 | 3.82 | Contralesional M1 |
| | | | | | 12.3, 13.8, 30.7 | 3.42 | Contralesional S1 |
| | | | | | 18.3, 12.8, 24.7 | 3.31 | Contralesional S1 |

Significant task relationship in the pre-lesion period were analyzed and indicated. On the x-axis, the lesioned side is indicated with a minus sign.
ACC anterior cingulate cortex, M1 primary motor cortex, S1 primary somatosensory cortex, SMA supplementary motor area.

**Table 2 Whole-brain statistical analysis of rCBF changes related to task condition: post-lesion period.**

**Post-lesion**

$\beta > 0$

| Cluster index | Voxels | Z-max | COG (x, y, z) (mm) | Region | Local maxima (x, y, z) (mm) | Z-value | Locus |
|---|---|---|---|---|---|---|---|
| 1 | 51363 | 6.48 | 4.92, −12.4, 20.3 | Contralesional V1/V2 Ipsilesional V3/IPS/MT/MST | 11.7, −18.2, 20.2 | 6.48 | Contralesional V1 |
| | | | | | 1.74, −17.7, 20.7 | 5.73 | Contralesional V1 |
| | | | | | 6.74, −15.7, 24.2 | 5.61 | Contralesional V2 |
| | | | | | 6.24, −18.7, 16.2 | 5.01 | Contralesional V2 |
| | | | | | 15.2, −12.2, 15.7 | 4.74 | Contralesional V1 |
| | | | | | 10.2, −10.7, 16.7 | 4.68 | Contralesional V2 |

$\beta < 0$

| Cluster index | Voxels | Z-max | COG (x, y, z) (mm) | Region | Local maxima (x, y, z) (mm) | Z-value | Locus |
|---|---|---|---|---|---|---|---|
| 3 | 11018 | 4.74 | −25.8, 19.8, 12.3 | Ipsilesional TE/S1/M1/PMv | −28.8, 16.3, 5.72 | 4.74 | Ipsilesional TE |
| | | | | | −25.3, 23.8, 0.2 | 3.63 | Ipsilesional TE |
| | | | | | −25.3, 24.3, 17.7 | 3.56 | Ipsilesional PMv |
| | | | | | −27.8, 16.8, 20.2 | 3.38 | Ipsilesional S1 |
| | | | | | −21.3, 20.3, 25.2 | 3.21 | Ipsilesional M1 |
| | | | | | −26.3, 31.3, 7.7 | 3.05 | Ipsilesional PMv |
| 2 | 9635 | 4.52 | 21.6, 21.3, 17.4 | Contralesional S1/M1/PMv | 20.2, 16.8, 18.7 | 4.52 | Contralesional S1 |
| | | | | | 23.7, 20.8, 17.7 | 4.06 | Contralesional S1 |
| | | | | | 19.7, 25.8, 13.7 | 3.75 | Contralesional PMv |
| | | | | | 21.7, 24.3, 18.2 | 3.55 | Contralesional PMv |
| | | | | | 23.2, 27.3, 15.7 | 3.49 | Contralesional PMv |
| | | | | | 18.7, 19.8, 25.7 | 3.36 | Contralesional M1 |
| 1 | 7515 | 4.73 | −27.7, −3.65, 9.99 | Ipsilesional V2/V3/V4/ TEO/TE | −27.8, −5.7, 9.7 | 4.73 | Ipsilesional V4 |
| | | | | | −28.8, −7.2, 16.2 | 3.55 | Ipsilesional V2 |
| | | | | | −31.8, 0.3, 13.2 | 3.53 | Ipsilesional V4 |
| | | | | | −29.8, 0.8, 9.2 | 3.38 | Ipsilesional TEO |
| | | | | | −31.8, 2.3, 2.2 | 2.94 | Ipsilesional TE |
| | | | | | −24.3, 3.8, 5.7 | 2.66 | Ipsilesional TE |

Same with Table 1 but in the post-lesion period.
IPS intraparietal sulcus area, M1 primary motor cortex, MST medial superior temporal area, MT middle temporal area, PMv ventral premotor area, S1 primary somatosensory cortex.

**Table 3 Whole-brain statistical analysis of rCBF changes related to the interaction between task condition and lesion.**

**Pre-post comparisons**

**Post > pre**

| Cluster index | Voxels | Z-max | COG (x, y, z) (mm) | Region | Local maxima (x, y, z) (mm) | Z-value | Locus |
|---|---|---|---|---|---|---|---|
| 14 | 7707 | 3.82 | 1.1, 12.1, 31.3 | MCC/SMA/S1 | 4.0, 15.5, 31.5 | 3.82 | Contralesional SMA |
| | | | | | 2.5, 16.0, 32.0 | 3.79 | Contralesional SMA |
| | | | | | −5.0, 11.5, 35.5 | 3.59 | Ipsilesional M1 |
| | | | | | 4.0, 12.0, 25.0 | 3.51 | Contralesional MCC |
| | | | | | −7.0, 6.5, 32.5 | 3.36 | Ipsilesional S1 |
| | | | | | −2.0, 9.5, 32.5 | 3.24 | Ipsilesional S1 |
| 13 | 4905 | 3.84 | −12.2, −0.9, 27.0 | Ipsilesional IPS/MT/MST/CD | −19.0, 1.0, 22.5 | 3.84 | Ipsilesional MST |
| | | | | | −14.5, −4.5, 30.0 | 3.68 | Ipsilesional MT |
| | | | | | −19.0, 0.0, 26.5 | 3.48 | Ipsilesional MST |
| | | | | | −14.5, 0.0, 28.5 | 3.44 | Ipsilesional LIP |
| | | | | | −7.5, −3.0, 31.0 | 3.15 | Ipsilesional LIP |
| | | | | | −12.0, 3.5, 20.5 | 3.13 | Ipsilesional CD |
| 12 | 4743 | 3.73 | 16.0, −2.0, 28.5 | Contralesional IPS/MT/MST | 14.5, −1.5, 32.5 | 3.73 | Contralesional LIP |
| | | | | | 21.0, 2.0, 28.0 | 3.57 | Contralesional TPJ |
| | | | | | 17.5, −5.0, 25.5 | 3.47 | Contralesional LIP |
| | | | | | 9.5, −3.5, 28.5 | 3.41 | Contralesional LIP |
| | | | | | 21.5, −0.5, 23.0 | 3.08 | Contralesional MST |
| | | | | | 15.5, −9.0, 27.5 | 2.96 | Contralesional LIP |
| 11 | 1236 | 3.06 | 1.9, 39.1, 27.8 | Contralesional dmPFC | 2.0, 38.0, 28.0 | 3.06 | Contralesional dmPFC |
| | | | | | 6.0, 37.5, 25 | 2.81 | Contralesional dmPFC |
| 10 | 735 | 3.33 | 1.2, −5.5, 19.4 | CB | 1.0, −5.5, 18.5 | 3.33 | Cerebellar vermis |
| 9 | 700 | 3.74 | 14.8, 37.3, 14.1 | Contralesional OFC | 14.5, 37.5, 13.5 | 3.74 | Contralesional OFC |
| 8 | 638 | 3.09 | −18.9, 11.5, 27.9 | Ipsilesional S1 | −19.0, 11.5, 27.5 | 3.09 | Ipsilesional S1 |
| 7 | 593 | 3.37 | 26.7, 14.4, 1.9 | Contralesional TE | 27.5, 14.5, 2.0 | 3.37 | Contralesional TE |
| 6 | 535 | 3.39 | −1.6, 50.5, 13.9 | Ipsilesional amPFC | −2.0, 52.0, 12.0 | 3.39 | Ipsilesional amPFC |
| | | | | | −2.0, 51.0, 14.5 | 3.36 | Ipsilesional amPFC |
| | | | | | −1.0, 51.5, 11.0 | 3.16 | Ipsilesional amPFC |
| 5 | 513 | 2.86 | 3.1, 31.5, 26.4 | Contralesional ACC | 2.0, 32.0, 25.5 | 2.86 | Contralesional ACC |
| | | | | | 4.5, 31.5, 27.5 | 2.82 | Contralesional ACC |
| 4 | 474 | 2.68 | 13.4, 3.4, 19.8 | Contralesional IPS | 12.5, 2.5, 22.5 | 2.68 | |
| | | | | | 14.0, 4.5, 15.5 | 2.64 | |
| | | | | | 9.0, 1.0, 25.5 | 2.33 | Contralesional LIP |
| 3 | 385 | 3.15 | 5.9, 24.9, 32.1 | Contralesional SMA | 6.0, 24.5, 32.5 | 3.15 | Contralesional SMA |
| 2 | 306 | 3.59 | 3.5, 9.0, −3.2 | Contralesional PN | 3.5, 9.0, −4.5 | 3.59 | Contralesional PN |
| 1 | 288 | 2.83 | 1.7, −9.7, 28.0 | V6 | 2.0, −9.5, 28.0 | 2.83 | Contralesional V6 |

**Pre > post**

| Cluster index | Voxels | Z-max | COG (x, y, z) (mm) | Region | Local maxima (x, y, z) (mm) | Z-value | Locus |
|---|---|---|---|---|---|---|---|
| 9 | 32893 | 5.88 | −11.6, −17.4, 19.7 | Ipsilesional V1/V2/V3v/V4v | −9.5, −20, 24.5 | 5.88 | Ipsilesional V1 |
| | | | | | −23.5, −7.0, 8.5 | 4.53 | Ipsilesional V3v |
| | | | | | −14.0, −9.0, 18.0 | 4.33 | Ipsilesional V1 |
| | | | | | −11.5, −16.0, 10.5 | 4.2 | Ipsilesional V2 |
| | | | | | −16.5, −20.0, 20.0 | 4.2 | Ipsilesional V1 |
| | | | | | −17.0, −14.5, 20.5 | 4.03 | Ipsilesional V1 |
| 8 | 2440 | 4.2 | 3.7, 9.1, 12.9 | Contralesional thalamus | 5.5, 8.5, 12.0 | 4.2 | Contralesional VPM |
| | | | | | 2.0, 12.5, 13.0 | 3.15 | Contralesional VL |
| 7 | 1056 | 3.67 | 4.4, 18.5, 6.2 | Contralesional OT | 4.0, 19.5, 8.0 | 3.67 | Contralesional OT |
| | | | | | 4.5, 18.0, 4.5 | 3.59 | Contralesional OT |
| 6 | 608 | 3.01 | 6.2, −16.5, 5.1 | Contralesional CB | 5.5, −17.0, 5.5 | 3.01 | Contralesional CB |
| | | | | | 6.5, −14.0, 0.5 | 2.38 | Contralesional CB |
| | | | | | 7.0, −14.0, −0.5 | 2.38 | Contralesional CB |
| 5 | 439 | 3.17 | 13.6, −11.1, 4.6 | Contralesional CB | 13.5, −10.5, 4.5 | 3.17 | Contralesional CB |
| 4 | 390 | 2.98 | 18.5, −13.5, 13.9 | Contralesional V2/V3 | 20.0, −14.5, 14.5 | 2.98 | Contralesional V2 |
| | | | | | 16.5, −11.5, 13.0 | 2.68 | Contralesional V2 |
| 3 | 365 | 2.87 | 1.6, 25.9, 15.7 | vmPFC/CD | 0.5, 26.0, 14.5 | 2.87 | Contralesional vmPFC |
| | | | | | 3.5, 24.0, 21.0 | 2.54 | Contralesional CD |
| | | | | | 3.5, 26.0 19.5 | 2.54 | Contralesional CD |
| | | | | | 1.0, 28.0, 10.0 | 2.53 | Contralesional vmPFC |
| | | | | | 0.5, 27.5, 11.0 | 2.52 | Contralesional vmPFC |
| 2 | 349 | 3.16 | −0.4, 4.4, 8.0 | Midbrain | −0.5, 4.5, 8.0 | 3.16 | midbrain |
| 1 | 323 | 2.78 | −2.8, −14.3, 6.0 | Ipsilesional CB | −3.0, −14.0, 6.0 | 2.78 | Ipsilesional CB |

Significant interaction between task and lesion (pre-lesion vs post-lesion) were analyzed separately and indicated. On the x-axis, the lesioned side is indicated with a minus sign.
*ACC* anterior cingulate cortex, *amPFC* anterior medial prefrontal cortex, *CB* cerebellum, *CD* caudate nucleus, *dmPFC* dorsomedial prefrontal cortex, *IPS* intraparietal sulcus area, *LIP* lateral intraparietal area, *M1* primary motor cortex, *MCC* midcingulate cortex, *MST* medial superior temporal area, *MT* middle temporal area, *OFC* orbitofrontal cortex, *OT* optic tract, *PN* pontine nuclei, *S1* primary somatosensory cortex, *SMA* supplementary motor area, *TPJ* temporo-parietal junction, *VL* ventral lateral nucleus, *vmPFC* ventromedial prefrontal cortex, *VPM* ventral posterior medial nucleus.

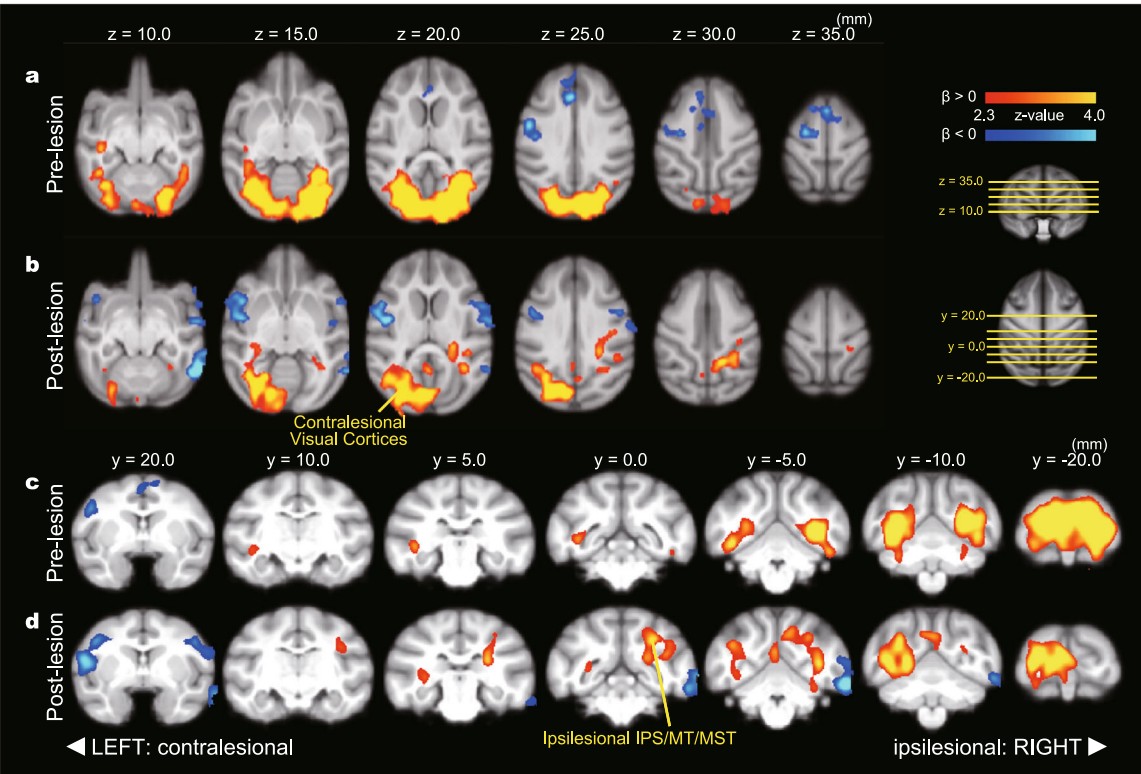

**Fig. 3 Task-dependent change in rCBF in pre- and post-lesion period.** Axial (**a**, **b**) and coronal (**c**, **d**) views of PET results obtained from monkeys C and T. The left and right sides of the brain were flipped to match the side of the lesion (the lesioned side is presented on the right side in this figure). **a**, **c** Brain areas with a significant relationship to task condition in the pre-lesion period. Red-yellow and blue-light blue show a significant positive and negative relationship to task condition, respectively. Only statistically significant clusters ($p < 0.05$) are shown. **b**, **d** Brain areas with a significant relationship to task condition in the post-lesion period. IPS intraparietal sulcus area, MCC midcingulate cortex, MST medial superior temporal area, MT middle temporal area, SMA supplementary motor area.

that the recording sites in the lbIPS overlapped with the LIP. To examine the causal contribution of the lbIPS to the visually guided saccades, reversible inactivation of the ipsi- or contralesional lbIPS was conducted with microinjection of muscimol (see Supplementary Table 1 and Supplementary Fig. 4 for details). Previous studies have shown that LIP inactivation does not affect simple visually guided saccades[22]. Figure 6a shows an example of ipsilesional lbIPS inactivation. The endpoints of saccades to different targets were intermingled in the affected field during inactivation. As the effects of muscimol injection varied for each target location, we selected representative target locations for which lbIPS inactivation resulted in the smallest (success rate [after] − success rate [before]) values from the affected and intact visual fields, respectively, and used the trials with target in those locations for further analysis (see "Methods" section). In most cases, there was an effect on the success rates of saccades toward the side contralateral to the injection (i.e., the affected visual field; Fig. 6b, one-tailed Fisher's exact test, $p < 0.05$, in 5/8 sessions, see Supplementary Table 1 for detail). In contrast, there was no change in the success rates of saccades toward the side ipsilateral to the injection (i.e., the intact visual field; one-tailed Fisher's exact test, $p > 0.05$, in 8/8 sessions, See Supplementary Table 1 for detail). Altogether, inactivation of the ipsilesional lbIPS impaired visually guided saccades toward the affected field, in which the RFs of ipsilesional lbIPS neurons were located (side contralateral to the injection).

Surprisingly, inactivation of the contralesional lbIPS caused different effects from ipsilesional lbIPS inactivation. As shown in Fig. 6c, d, contralesional lbIPS inactivation did not impair the

visually guided saccades toward the side contralateral to the injection (i.e., the intact visual field; one-sided Fisher's exact test, $p > 0.05$, in 8/8 sessions). However, there was a reduction in the success rate of saccades toward the side ipsilateral to the injection (i.e., the affected visual field) in many cases (Fig. 6d, one-sided Fisher's exact test, $p < 0.05$, in 5/8 sessions).

These results support and strengthen the hypothesis that the bilateral IPS areas contribute to residual visuomotor function after V1 lesion, as suggested by the PET study. First, the ipsilesional lbIPS, differently from intact animals, is involved in the control of simple visually guided saccades toward the affected field. Second, the contralesional lbIPS also controls saccades toward the affected field. This was unexpected, because, as we confirmed by single-unit recordings, neurons in the contralesional lbIPS responded primarily to targets in the intact field.

To quantify further the behavioral changes after lbIPS inactivation, we separated the saccade endpoint errors into direction and eccentricity errors, and analyzed their distribution in each session. As shown in Fig. 7, the distributions of saccade endpoints during lbIPS inactivation were not unimodal, but multimodal with separate peaks, one peak around target (on-target saccades), others to nontarget locations (off-target saccades), in some cases. Notably, error saccades were often made toward specific locations outside the target window in these examples. In monkey C, error saccades were often directed to the neighboring possible target locations relative to the actual target (Fig. 7a, b), which was supported by the multimodality of the distribution of endpoint directions detected by Silverman's test (Fig. 7b and Supplementary Fig. 7). In monkey U, error saccades

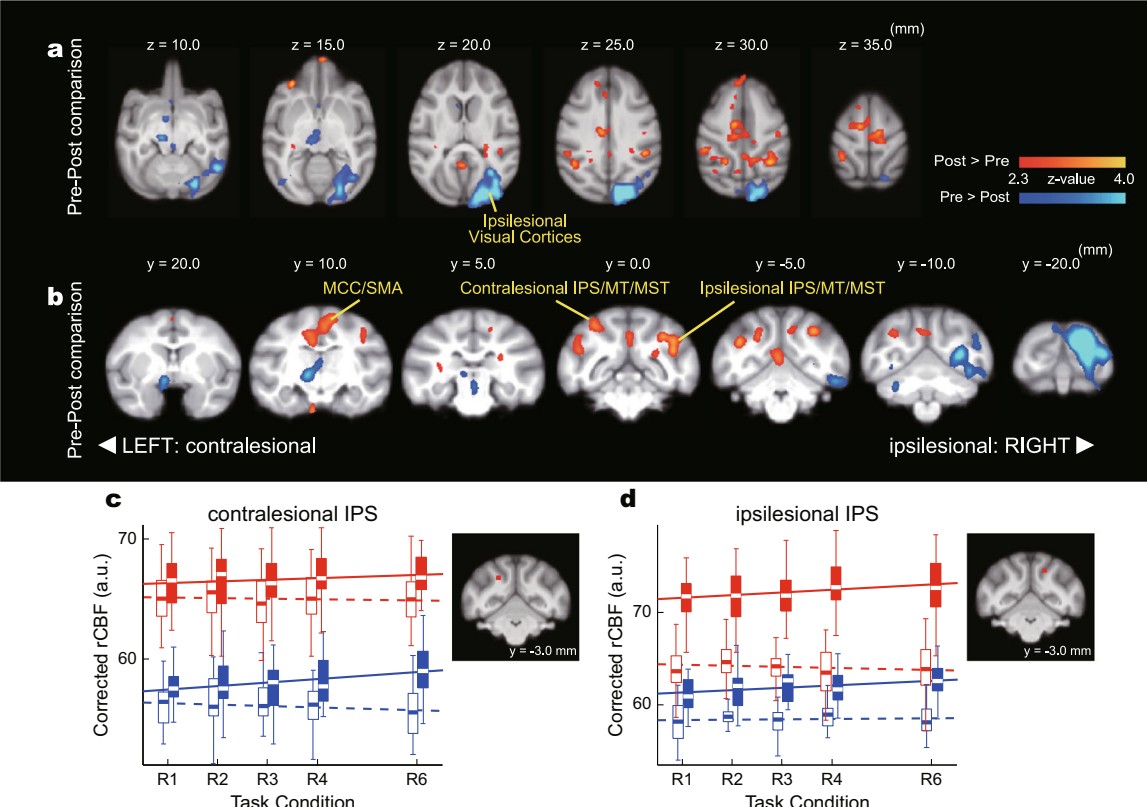

**Fig. 4 Interaction effects on rCBF between task and lesion.** Interaction effects of task and lesion showing the change in task relation between pre- and post-lesion period. **a**, **b** Brain areas with a significant interaction effect between task condition and lesion. Red-yellow and blue-light blue show a significant increase and decrease of task relationship in the post-lesion period compared to the pre-lesion period, respectively. **c**, **d** The relationship between rCBF (corrected for global signal) and task condition in the contra- and ipsilesional IPS, respectively (regions of Interests were 5 × 5 × 5 voxels around local maxima of post > pre contrast; ipsilesional IPS: $x = −7.5$, $y = −3.0$, $z = 31.0$ mm; contralesional IPS: $x = 9.5$, $y = −3.5$, $z = 28.5$ mm; red squares in each inset). Box plots and regression lines are shown. Blue open boxes with a dotted line indicate monkey C, pre-lesion data. Red open boxes with a dotted line indicate monkey T, pre-lesion data. Blue filled boxes with a continuous blue line indicate monkey C, post-lesion data. Red filled boxes with a continuous line indicate monkey T, post-lesion data. Regression coefficients: ipsilesional IPS in pre-lesion period, monkey C = 0.04, monkey T = −0.12; ipsilesional IPS in post-lesion period, monkey C = 0.26, monkey T = 0.29; contralesional IPS in pre-lesion period, monkey C = −0.12, monkey T = −0.04; contralesional IPS in post-lesion period, monkey C = 0.29, monkey T = 0.14.

frequently landed on a particular area that was different from any possible target location (upper left visual field, eccentricity = ~9°, ipsilesional lbIPS: Fig. 7d and Supplementary Fig. 7), suggesting that monkey U had a habit of making saccades to that area. Such multimodal distribution of saccade direction was observed in 2/8 and 3/8 cases of ipsi- and contralesional lbIPS inactivation, respectively (asterisks in Supplementary Fig. 7). In addition, a multimodal distribution of saccade eccentricity was observed in two sessions (0/8 and 2/8 cases of ipsi- and contralesional lbIPS inactivation, respectively; Supplementary Fig. 8).

This multimodality may not be explained merely by deficits in the saccade motor system. Direction and eccentricity errors of on-target saccades significantly increased from the control condition (Supplementary Tables 2 and 3). These effects on on-target saccades could be explained by deficits in either visual or saccade motor system. Moreover, we observed increase in frequency of off-target saccades during lbIPS inactivation (Fig. 7 and Supplementary Fig. 5). In the latter case, we suggest that the saccades were directed by internal information—by previous knowledge about possible target locations in monkey C or by habit to make saccades to a certain location in monkey U. In one session, the error saccades were directed to a location symmetrical to the target in the affected field (Supplementary Fig. 5, ContU2), which might be related to the previous report in human

blindsight[23] and may be due to the deficit in visuomotor transformation. In summary, the saccades became less accurate after the V1 lesion (before the lbIPS inactivation, Supplementary Fig. 2), as confirmed by comparison of saccades in the control sessions for the inactivation experiments (see Supplementary Fig. 9). However, we observed increase in direction and eccentricity errors of on-target saccades (Supplementary Tables 2 and 3) and frequency of off-target saccades during lbIPS inactivation (Fig. 7 and Supplementary Fig. 5). Thus, the effect of lbIPS inactivation was significant and the results cannot be solely explained by the effect of V1 lesion.

After lbIPS inactivation, an increase in saccadic reaction time was observed in the affected field in 3 out of 16 sessions (all the data including error saccades: $p < 0.05$ by the Wilcoxon rank-sum test, see Supplementary Table 1). Saccadic reaction time toward the intact field changed in many sessions; however, the directions of the changes were inconsistent. Overall, the effects of lbIPS inactivation on reaction times were minor.

**Other regions of interests.** Here, we focused on the bilateral lbIPS since they are important in visuomotor function in normal monkeys[24], and our whole-brain analysis showed a significant change in these regions after V1 lesion. However, there were other regions

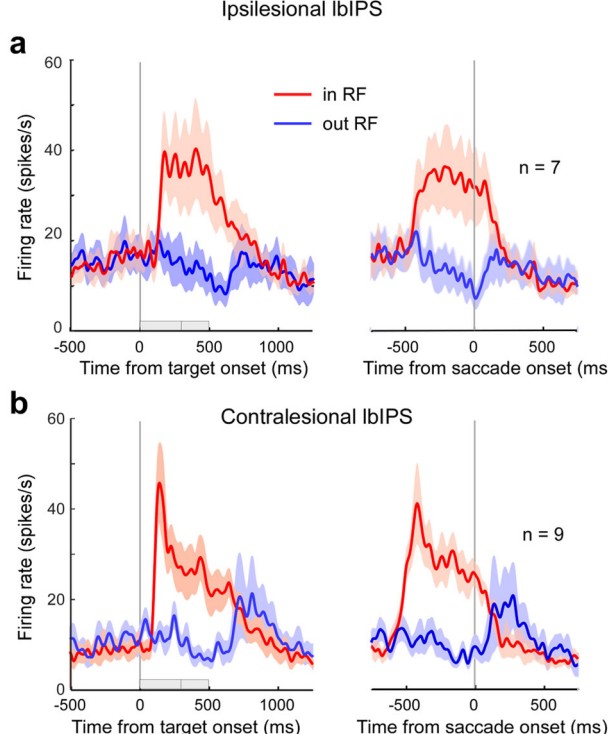

**Fig. 5 Responses of lbIPS neurons during the overlap saccade task.** Population activity of neurons in the ipsilesional (**a**) and contralesional (**b**) lbIPS during the overlap visually guided saccade task. The left and right panels are aligned to target onset and saccade onset (vertical gray line), respectively. The red traces indicate the mean firing rate when the target was presented in the RF. Blue traces indicate the mean firing rate when the target was presented outside the RF. Shaded areas indicate SEM. Gray bars on the horizontal axis of the left panels indicate the overlap periods of the FP and target (varying between 300 and 500 ms).

that should be mentioned. The bilateral MT and MST areas and ipsilesional caudate nucleus showed a significant change between the pre- and post-lesion periods, and formed clusters with the IPS areas (Table 3). Similarly, rCBF in the midcingulate cortex (MCC) showed a significant change in the post-lesion period compared to the pre-lesion period, suggesting the involvement of MCC in residual visuomotor function. The frontal eye field (FEF) and SC, along with the LIP, are known as critical regions for visuomotor processing in normal subjects[25]. However, we could not find a significant task relationship in the FEF or SC in the current experiment. Detailed individual analysis (Supplementary Fig. 3) revealed that monkey C showed a consistent task relationship to the bilateral FEF in the pre-lesion period (right: $Z$-max = 3.82 at $x = -14.0$, $y = 32.5$, $z = 21.0$; left: $Z$-max = 3.36 at $x = 19.5$, $y = 32.5$, $z = 23.0$) and a slightly stronger task relationship in the post-lesion period (right, ipsilesional: $Z$-max = 4.20 at $x = -14.0$, $y = 33.0$, $z = 21.5$; left, contralesional: $Z$-max = 5.41 at $x = 15.0$, $y = 33.0$, $z = 23.0$), which failed to reach statistical significance as a cluster. Monkey T did not show any task relationship in the FEF or dorsolateral prefrontal cortex, except for the ipsilesional FEF/dorsolateral prefrontal cortex in the post-lesion period ($Z$-max = 3.23 at $x = -15.0$, $y = 26.5$, $z = 23.0$). The SC was even less clear; only the ipsilesional SC showed a weak relationship to task condition in the post-lesion period in monkey C ($Z$-max = 2.63 at $x = -4.5$, $y = 6.5$, $z = 13.5$), which was not surprising considering the limited size of SC. According to previous reports, we believe that the FEF and SC are involved in visuomotor processing during the pre- and post-lesion periods[5,15,18,26], and further studies are needed to

clarify their possible contributions to residual visuomotor function. In addition, to search regions which showed altered task- and lesion-dependent connectivity with IPS areas, we analyzed psychophysiological interactions (PPI)[27,28], using ipsi- and contralesional IPS seeds (same ROIs as in Fig. 4d, e). The results showed significant increase in PPI in the post-lesion period compared to the pre-lesion period in MCC, contralesional midbrain, retrosplenial cortex, and contralesional primary motor cortex, with both ipsi- and contralesional IPS areas (Supplementary Fig. 10 and Supplementary Table 4). The results suggest both ipsi- and contralesional IPS areas changed their functional connectivity after lesioning, which might help residual visuomotor ability.

## Discussion

Whole-brain PET analysis revealed the regions related to visuomotor processing by detecting the voxels, whose rCBF was changed proportionally to the number of visuomotor events. We found that the bilateral IPS areas in the posterior parietal cortex showed significantly increased relationships to visuomotor events after V1 lesion, suggesting that these areas become more critically involved in visuomotor function in blindsight (Fig. 4). Because of its visuomotor function in intact monkeys[20,21,24], we decided to focus on the lbIPS. Single-unit recordings from the bilateral lbIPSs showed that the neural responses were very similar between the ipsi- and contralesional lbIPSs, which share some properties with LIP neurons of intact monkeys in previous reports (Fig. 5). As described in "Results", the inactivation areas were presumed to overlap with the LIP, a saccadic visuomotor region in the posterior parietal cortex (Supplementary Fig. 4). However, lbIPS inactivation in V1-lesioned animals showed completely different results from previous reports in intact monkeys. lbIPS inactivation has very little effect on simple visually guided saccades in intact monkeys[22,29], whereas ipsi- and contralesional lbIPS inactivation significantly impaired visually guided saccades toward targets in the affected field after V1 lesion (Fig. 6). This result indicates that the importance of the lbIPS in visuomotor function increased after V1 lesion, which is in accordance with our PET results. Another important finding here is that both ipsi- and contralesional lbIPS were involved in the residual visuomotor function in the affected field. Most lbIPS neurons have a RF in their contralateral visual field[21]. However, inactivation of the contralesional lbIPS did not impair visuomotor behavior in the intact field, but significantly impaired it in the affected field. The overall results clearly demonstrated functional changes in the lbIPS after V1 lesion, and indicate that both ipsi- and contralesional lbIPS play critical roles in residual visuomotor function in blindsight. In the following sections, we will discuss the importance of the lbIPS in blindsight and possible neural mechanisms.

Involvement of ipsilesional IPS area in visuomotor function after V1 lesion suggested by PET imaging was supported by unit recordings, where ipsilesional lbIPS neurons showed clear phasic visual responses similarly to contralesional/intact lbIPS neurons. As for the thalamic relay of visual information to the cortex, the pulvinar[30–33], and/or the lateral geniculate nucleus[34–36] are considered to contribute to blindsight, presumably depending on the properties of visual signals to be processed, as proposed by Tamietto and Morrone[37]. Area MT is thought to receive visual inputs from these subcortical areas[31,35] and provide visual information to the lbIPS and MST regions in blindsight; thus, it is reasonable that we observed a cluster in the ipsilesional IPS/MT/MST regions in our imaging results. However, further studies are needed to understand the functional changes in early visual processing to compensate for the loss of V1.

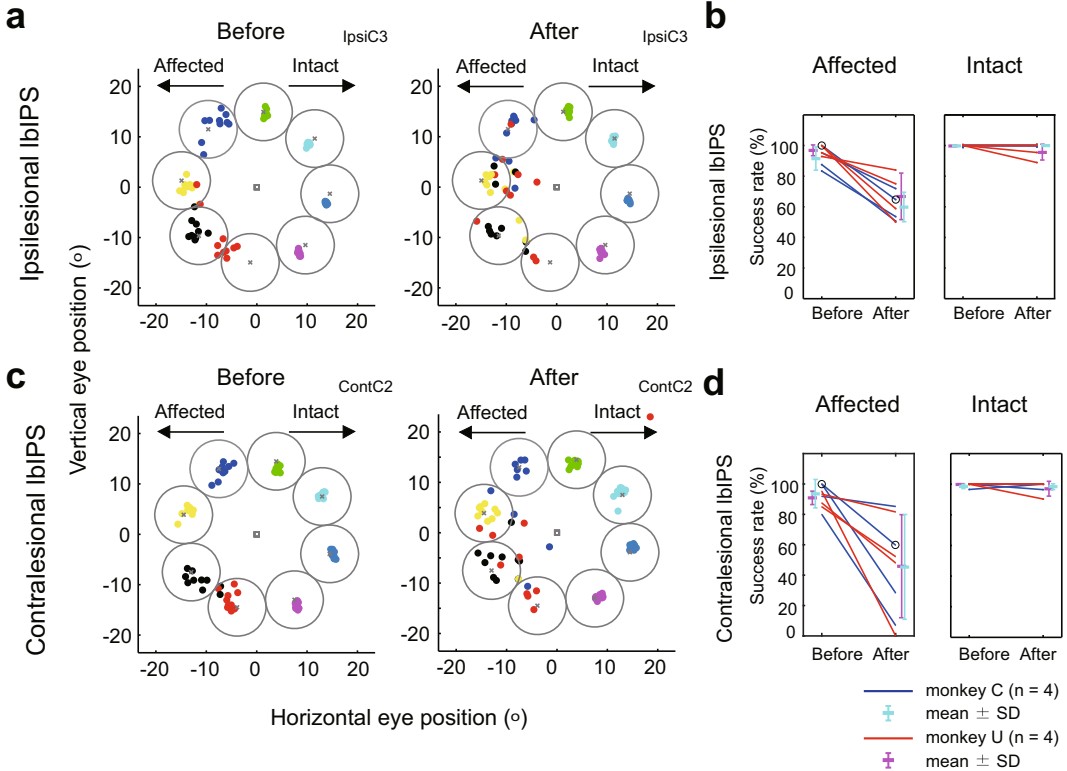

**Fig. 6 Effects of reversible inactivation of the lbIPS in the step saccade task. a**, **c** Each dot represents saccade endpoints and is color coded by target location. **a** Before (left) and after (right) inactivation of the ipsilesional lbIPS. **b** Before (left) and after (right) inactivation of the contralesional lbIPS. Target windows for successful saccades are indicated as circles. **b**, **d** Success rates of visually guided saccades to the target in the affected (left) and intact (right) visual field. Inactivation of the ipsilesional (**b**) and contralesional (**d**) lbIPS. The data of target locations with the smallest (success rate [after] − success rate [before]) values by inactivation are plotted in **b** and **d**. The data for saccades toward the affected and intact visual hemifield are represented in the left and right panel, respectively. The data points of individual experiments, before injection (before) and after injection (after), are presented on the left and right side of each panel with connecting lines, respectively. Experimental sessions in **a** and **c** are marked by open circles in **b** and **d**. For session IDs (IpsiC3 and ContC2), see Supplementary Table 1. The mean and SD of success rates in each monkey are plotted in cyan (monkey C) and magenta (magenta: monkey U). The target window (gray circle) was a circle with a radius of half the distance between target locations (radius = eccentricity × sin[45/2]). Thus, these windows did not overlap with each other.

Previous studies using neuroimaging techniques reported that the ipsilesional IPS area shows a blood flow response to a visual stimulus in the affected field in V1-lesioned subjects[17]. We confirmed this observation with single-unit recordings. Ipsilesional lbIPS neurons retained many characteristics of those in intact monkeys, i.e., phasic visual responses, sustained activity during the delay period, and perisaccadic responses to a target in the affected field. However, the latency of the phasic visual response was longer in the ipsilesional lbIPS (110–170 ms) compared to those in intact monkeys and contralesional lbIPS (~110 ms). A similar response delay was observed in area MT after V1 lesion[16]. On the other hand, the ipsilesional lbIPS did not decrease visual response magnitude (Fig. 5 and Supplementary Fig. 5), which could be explained as follows. One possibility is that the phasic response was enhanced to compensate for the reduced visual input and to improve the signal-to-noise ratio. This may be due to a plastic change in the bottom-up visual pathway during the recovery period or in top-down attention, which is observed frequently in various brain regions[38]. The other possibility is the facilitation of baseline activity by general activation or disinhibition. Our single-unit recordings showed a slight increase in the baseline firing of the ipsilesional lbIPS compared to the contralesional lbIPS; the difference was not statistically significant, but may support the latter hypothesis. In intact monkeys, the LIP is more involved in the target detection/selection from a complex background rather than a simple reflexive saccade as

shown in neural activity[24,39,40] and reversible inactivation experiment[22]. After V1 lesion, visual information becomes more ambiguous and less reliable, which makes a simple saccade task more complex and difficult like search task with distractors and a double-target task in the intact animals, which might require the LIP. Our overall results suggest that the ipsilesional lbIPS compensates for the reduced visual input by adaptation, although further studies are required to understand the underlying neural mechanisms.

The most unexpected finding here was that inactivation of the contralesional lbIPS also impaired visually guided saccades toward the affected field, although saccades to the intact hemifield were not affected (Fig. 6). The results strongly suggest that both the ipsilesional and contralesional lbIPS are involved in residual visuomotor function in blindsight, which is counterintuitive as the RFs of most lbIPS neurons are in their contralateral field[41]. One possible explanation would be that some contralesional lbIPS neurons were involved in visuomotor processing for saccade execution through response to the target in the affected field. Some LIP neurons reportedly extend their RF to the ipsilateral visual hemifield[21]. In addition, a tractography study showed that a blindsight patient possessed an ascending pathway from the ipsilesional SC to the contralesional LIP, which is absent in healthy subjects[42]. Thus, it is possible that some neurons in the contralesional lbIPS are involved in visuomotor function for saccades toward the affected field. However, if so, either

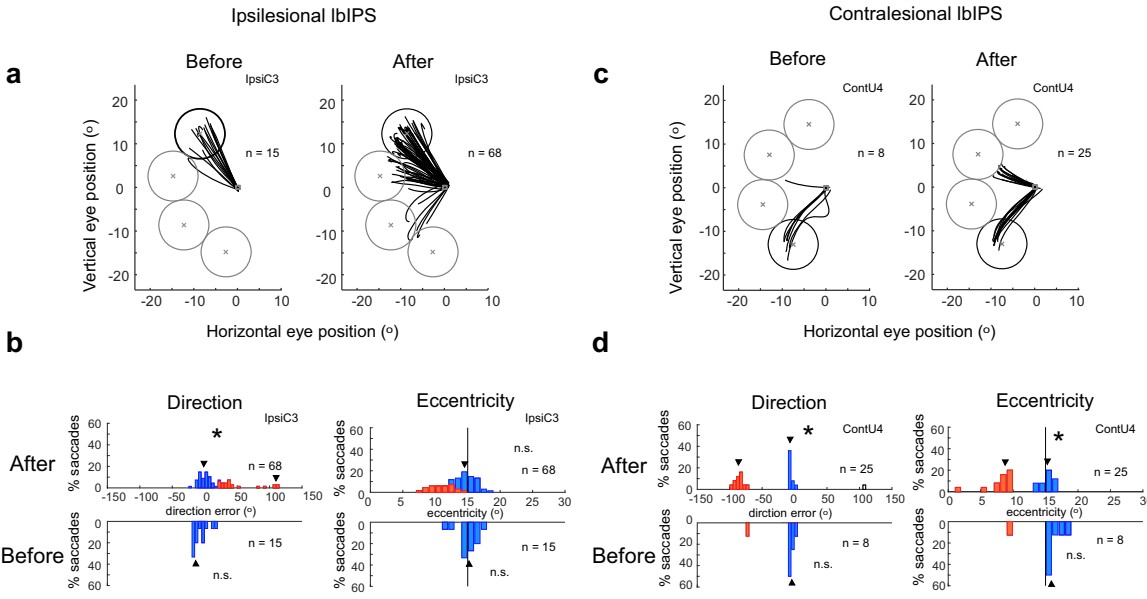

**Fig. 7 Saccade trajectories and endpoints in example sessions before and after lbIPS inactivation. a, b** Saccade trajectories and the distribution of saccade endpoints (direction and eccentricity) in a single session (monkey C, IpsiC3). **c, d** Saccade trajectories and the distribution of saccade endpoints (direction and eccentricity) in a single session (monkey U, contU4). Circles in **a** and **b** indicate target windows (the circle with the thick line indicates target location). **b, d** The distribution of error angles in direction (left) and eccentricity (right) before (lower) and after (upper) lbIPS inactivation. The multimodality of distribution evaluated by Silverman's test ($p < 0.05$) is indicated by an asterisk in each panel. Arrowheads indicate local modes estimated by Silverman's test. Blue and red columns indicate correct saccades and error saccades, respectively. An open column indicates the saccades removed from the analysis as outliers. For session IDs (IpsiC3 and ContC2), see Supplementary Table 1.

ipsilesional or contralesional lbIPS inactivation would yield weak deficits because the other lbIPS can control saccades to the affected field, which was not the case in our experiments. In addition, there were no such contralesional lbIPS neurons showing a clear response to stimuli in the affected field in the limited sample we examined ($n = 9$). Another possibility is that the contralesional lbIPS contributed to visuomotor processing for saccade execution through interhemispheric interactions as suggested for human blindsight subject[43]. Several studies have suggested interhemispheric interactions between bilateral LIPs. For example, Balan et al. showed that inactivation of the unilateral LIP resulted in an increased BOLD response in the contralateral LIP[44]. We found that some contralesional lbIPS neurons decreased activity during the delay period to the targets outside their RF (Fig. 5 and Supplementary Fig. 6), which may be related to interhemispheric inhibition. Alternatively, the contralesional lbIPS might support the ipsilesional lbIPS by lowering the threshold through a facilitatory/disinhibitory mechanism, as discussed in the previous section. In either way, we think that the bilateral lbIPSs interact with each other for the residual visuomotor function.

Whole-brain analysis using PET revealed several brain regions that showed significant changes in visuomotor-related rCBF by V1 lesion. Although we focused on the bilateral lbIPSs in this study, there are many other regions that are potentially involved in residual visuomotor function. The bilateral MT and MST areas showed significant changes after V1 lesion, forming a cluster with the IPS area. This cluster also included a part of the caudate nucleus, which may be involved in residual visuomotor function or behavioral adaptation through corticostriatal interactions[45]. We also found a change in the visuomotor activity in the MCC after V1 lesion. The MCC has anatomical connections with the FEF[46] and supplementary eye field[47], and is considered to be involved in saccade control[48,49]. The MCC is also known to receive inputs from the parietal cortex[50], which suggests that IPS–MCC interactions may also be involved in residual

visuomotor function. Indeed, our PPI analysis showed changes in task-dependent connectivity between MCC and bilateral IPS areas in post-lesion period (Supplementary Fig. 10), suggesting that MCC–IPS interactions mediated the residual visuomotor function. Conversely, we could not find any consistent change in other visuomotor-related regions, such as the FEF, supplementary eye field, or SC, even though we are sure that the SC has a crucial role in residual visuomotor function[5,18]. As these regions are finely organized in a retinotopic manner, the current task may activate only a limited portion of neurons, which might have made it difficult for us to detect changes in activity by PET. However, we could see significant functional connectivity between the contralesional midbrain and bilateral IPS areas in both monkeys, and visuomotor-related activity in the dorsolateral prefrontal cortex and FEF in one of the monkeys. Considering the known anatomical connections from the IPS and MCC, it is possible that the SC, FEF, and dorsolateral prefrontal cortex are also involved in residual visuomotor function in more spatially selective manner.

By combining whole-brain neuroimaging analysis and neurophysiological methods, we found that the bilateral lbIPSs play a critical role in residual visuomotor function in blindsight. The bilateral lbIPSs showed a causal contribution to the control of eye movements in blindsight, which could be observed only in an active behavioral context. Altogether, the present study clarified the novel functions of the lbIPS, which overlaps with LIP, in blindsight. This might have been acquired during the recovery process from V1 lesion for the direct control of behavior. The posterior parietal cortex, such as IPS areas, which is involved in more demanding tasks in intact animals, could be recruited for the control of a simple visually guided saccade task with limited visual information after V1 lesion. The present results could be explained by either the enhancement of the top-down modulation of visuomotor processing at the lower level or the upregulation of bottom-up visual processing through plastic changes in the residual visual pathways spared by V1 lesion, including those

mediated by the SC. Further studies to analyze neuronal activity in these regions and the effects of region/pathway-selective manipulation spanning these areas would give us clues for a further understanding of the neural mechanism of blindsight.

## Methods

**Animals and surgery.** Three macaque monkeys (monkey C, *Macaca mulatta*, 6.5 kg male; monkey T, *M. mulatta*, 8.4 kg male; monkey U, *Macaca fuscata*, female 5.6 kg) were used in this study. Monkey C (6–8 months after right V1 lesion) and monkey T (6–8 months after left V1 lesion) were used for PET experiments. Monkeys C and U were used for neurophysiological experiments at 17–18 months and 96–103 months after right V1 lesion, respectively. All experimental procedures were performed in accordance with the National Institutes of Health Guidelines for the Care and Use of Laboratory Animals and Basic Policies for the Conduct of Animals Experiments in Research Institutions by MEXT, Japan, and approved by the Committee for Animal Experiments at the National Institutes of Natural Sciences, Japan and the Central Research Laboratory in Hamamatsu Photonics, Hamamatsu, Japan, and the Ethics Committee on Animal Care and Use of RIKEN Center for Life Science Technologies, Japan. The monkeys were purchased from the National Bioresource Project by MEXT (monkey U) or a domestic breeder (monkeys C and T) in Japan. All surgeries were performed in aseptic conditions under isoflurane anesthesia (1.0–1.5%). A head holder was implanted to stabilize the head position of all monkeys. For monkey U, an eye coil was also implanted in one eye to monitor eye position.

**Recording and analysis of eye movements.** Eye positions were recorded by an infrared video-based system (monkeys T and C, 250-Hz sampling rate; Hamamatsu Photonics, Japan) or a magnetic search coil method (monkey U, 1-kHz sampling rate; Enzanshi Kogyo, Japan). Visual stimuli were presented on an LCD display positioned at 24 or 42 cm in front of the monkeys' eyes. The monkeys sat on a monkey chair with their head fixed to the chair, using a head holder implanted on their skulls.

For the PET experiments, eye velocity was calculated by an eight-point derivative algorithm[51]. Saccades were detected when eye velocity exceeded 200°/s. Then, saccade onset was defined as the time when velocity exceeded 30°/s before reaching 200°/s, and saccade offset was defined as the time when velocity declined <30°/s for three consecutive data points. Due to the limited sampling rate and low signal-to-noise ratio, we could not obtain reliable peak velocity information. Saccadic reaction times and saccade accuracy (location of the saccade end point relative to target location in the horizontal direction) during PET scanning were compared by two-way analysis of variance (pre/post-lesion and rightward/leftward saccade) on ranks of the data for each monkey, which was followed by a post hoc Mann–Whitney test with Bonferroni's correction. Confidence intervals of the interquartile range were estimated by the bootstrap method with 2000 iterations.

For the physiological experiments, the eye positions of monkey C were first converted to 1 kHz by a second-order Butterworth low-pass filter with a 44-Hz cutoff frequency, to match those of monkey U. Saccades were detected when eye velocity exceeded 100°/s. Saccade onset was defined as the time when velocity exceeded 30°/s before reaching 100°/s, and saccade offset was defined as the time when velocity went <30°/s.

**Visually guided saccade tasks.** In this study, we used three different visually guided saccade tasks. (1) A round saccade task was used for the PET experiments (monkeys T and C). (2) A step saccade task was used in the muscimol injection experiments and in single-unit recordings from the lbIPS after V1 lesion (monkey C and U). This task was also used for the initial training of the monkeys. (3) An overlap saccade task was used for single-unit recordings from the lbIPS after V1 lesion (monkey U). Visual display and data storage were controlled by a real-time experimental control system (TEMPO for Windows; Reflective Computing). Fixation points (FPs) were set to a diameter of 0.4° with a luminance of 31.2 cd/m² against a background of 2.4 cd/m² (monkeys T and C) or a diameter of 0.7° with a luminance of 49.1 cd/m² against a background of 1.4 cd/m² (monkey U). In most cases, the target was of the same size and luminance as the FP (the target was slightly larger than the FP [0.5° in diameter] for monkeys T and C when the target was presented peripherally). If the monkey performed appropriate saccade(s) with a certain temporal/spatial accuracy defined by each task and obtained a juice reward, that trial was considered as a success.

Round saccade task: the monkeys were required to make serial visually guided saccades in each trial (Fig. 2). A trial started with the appearance of a central target on which the monkeys had to fixate. After a certain period of fixation, the central target disappeared and a peripheral target was presented simultaneously either right or left, 10° distant from the central target. The monkeys were required to make a centrifugal saccade to the peripheral target within 500 ms and maintain fixation on it for an additional time period, followed by a return centripetal saccade to the central target when the peripheral target disappeared and the central target reappeared. This pair of centrifugal and centripetal saccades was regarded as a "round," and the monkeys were required to make 1, 2, 3, 4, or 6 consecutive rounds in a trial to obtain a juice reward. The direction of the first saccade in a trial was pseudorandomized, but the direction in the following round (if any) changed

alternately. The duration of fixation was modulated depending on the task condition to make the time required to finish a single trial constant across task conditions, so that the number of saccades and amount of reward given in each scanning session were controlled (Fig. 2b).

Step saccade task: after fixating on the central FP for 800–1600 ms, the FP was turned off and a peripheral target was presented simultaneously. The monkeys were required to move their eyes to the target within 1 s and reach the target location within 200 ms after leaving the FP to receive a drop of juice reward. During the training sessions, four or eight possible target locations with the same eccentricity were used. We then used eight possible target locations with the same eccentricity, separated by 45° from each other, for the muscimol injection experiments.

Overlap saccade task: In the single-unit recording experiments in monkey U, an overlap saccade task was used. After fixating on the central FP for 800–1600 ms, a peripheral target was presented. Unlike the step saccade task, the monkey was required to maintain fixation for an additional delay period (300, 400, or 500 ms: varied in each recording session) until the FP disappeared. Then, the monkey was required to move its eyes within 1 s and reach the target location within 200 ms after leaving the FP to receive a juice reward. There were ten possible target locations with the same eccentricity (5 directions of upper 60°, upper 30°, 0°, lower 30°, and lower 60° in the hemi-visual field).

**PET scan and data analysis.** After sufficient training on the step and round saccade tasks, PET scans were performed on monkeys C and T. Twenty slices with a center-to-center distance of 3.6 mm were collected simultaneously by an SHR7700 camera (Hamamatsu Photonics K.K., Japan) in 3D mode. The transaxial resolution of the PET scanner was 2.6 mm at full-width half-maximum. The scan was initiated automatically when the radioactivity in the brain became >30 kcps after the delivery of [¹⁵O]H₂O (~300 MBq in 1.5 mL) via a cannula placed in the sural vein. The monkeys started the round saccade task preceding the start of the scan and continued for ~20 trials, which took ~2 min. Task condition (number of rounds in a trial) was fixed in a single scan. Five scans (once for each task condition, randomized in order) were repeated four times, interposing a scan with no task (resting condition), in 1 day with an inter-scan interval of ~15 min. The PET experiments were conducted twice a week with at least a 48-h interval. The 3D emission PET data was reconstructed with a filtered back projection to obtain images, which represent a relative spatial distribution of rCBF.

Data obtained from PET scanning sessions in which the subjects showed poor performance (monkey T: >3 error trials or <13 correct trials in a scan time window, 27/415; monkey C: >2 error trials or <12 correct trials in a scan time window, 43/357) were excluded from the analysis. Also, sessions in which the global count was significantly high or low (>3 SD) were discarded (monkey T: 4/415; monkey C: 1/357).

Spatial preprocessing and statistical analysis were performed by using FSL 5.0.9 (FMRIB Software Library, Oxford University). The rCBF PET images were merged, motion corrected within an either period of pre- and post-V1 lesion, and co-registered to the corresponding structural T1-weighted magnetic resonance image (T1w-MRI) at pre- and post-V1 lesion, respectively, with a linear registration using FLIRT. The T1w-MRI was used to determine a nonlinear warpfield using FNIRT to a standard anterior–posterior commissure space of macaque brain[52], which was then applied to all the PET data to embed in the standard space with a matrix of 187 × 195 × 120 and a voxel size of 0.5 × 0.5 × 0.5 mm. For the post-V1 lesion data, the lesion mask was created using MRIs and used for unmasking when determining warpfield to avoid distortion in the lesion area. The PET data was spatially smoothed with a Gaussian filter of 4 mm at full-width half-maximum and masked to include only brain region. Then, the data from monkey C were right-and-left flipped to match the side of the lesion with that of monkey T. To identify brain areas that showed a common relationship to the task condition in both monkeys, first-level statistical analyses were performed using FEAT[53,54]. Regional rCBFs were regressed by subject, task condition (1, 2, 3, 4, or 6) in the pre- and post-lesion periods separately. We also used a regressor for global signal to remove its effect. The change in task relationship between the pre- and post-lesion periods was tested by higher-level fixed effect analysis. The statistical threshold was set at Z > 2.3, and each cluster of voxels with a higher Z-value was tested using Gaussian random field theory (p < 0.05).

**Unilateral V1 lesion.** After several months of pre-lesional training (described above), the left V1 of one monkey (monkey T) and right V1 of two monkeys (monkeys C and U) were surgically removed by aspiration with a suction tube, as described previously[8] (Fig. 1). The lesion extent was confirmed on the postmortem brain fixed with 4% paraformaldehyde in Nissl-stained parasagittal sections for monkey T and with T1 imaging using 7 T MRI (Siemens) for monkeys C and U (Supplementary Fig. 1). The opercular surface of V1 was removed, except for its ventrolateral part representing the foveal region. The posterior half of the calcarine fissure was also aspirated. Thus, the cortical area representing 2–30° eccentricities in the contralateral visual field was removed.

**Histological analysis.** The formalin-fixed brains of the three monkeys were processed to delineate the extent of V1 lesion. The brain of monkey T was sectioned in 50 μm thickness along the parasagittal axis and Nissl stained. The brains of the two macaques (C and U) were sectioned for the occipital lobes and placed in sample

containers filled with Fluorinert. They were scanned with a 7 T MRI scanner (Magnetom 7 T, Erlangen, Germany) with a knee and insert coils using a 3D gradient echo sequence (TR/TE = 35/10 ms, FA 20° in isotropic 150 μm).

**Single-unit recording and analysis of neural data.** In monkey U, the chambers for single-unit recordings were fixed on the skull and arranged for inserting electrodes into the lbIPS through the guide tubes. Tungsten microelectrodes (FHC: 1.2–1.7 MΩ) were used for recording the single-unit activity of lbIPS neurons. Recording sites were estimated based on the relative positions between the lbIPS and the recording chamber on MRIs (Supplementary Fig. 4). Neurons with responses to the visual stimuli during the visually guided saccade task and with sampled trial numbers larger than seven for each of the targets in and outside the RF were served for the analysis. They were mainly in the caudal part of the lbIPS. The recordings were made primarily at 5–10 mm from the cortical surface along the sulcus. Spikes were detected by a spike sorting system (ASD; AlphaOmega) with a 1-kHz sampling rate. To identify the visual receptive field of each neuron, we compared the neural responses to a total of 40 target locations in the visual field. Ten of them were located on the same eccentricity from the central FP (5, 10, 15, and 20°, respectively). The center of the RF was defined as the point at which target presentation induced the maximum average firing rate during 50–250 ms after target presentation and used for further data analysis (Fig. 5). Target location outside the RF used for data analysis (Fig. 5) was defined as the point symmetry location with the target in the RF. The spike data were converted to spike density functions with a Gaussian kernel with a sigma of 15 ms. Each recorded neuron was first tested to see if it showed clear visual responses. The control period activity was measured as the average firing rate between 200 ms before and 40 ms after target onset (control period 1). Control period 1 included 40 ms after target onset to avoid misidentification of the visual response latency in a neuron showing buildup activity around target onset. Using a 25-ms sliding window (1-ms step), we defined visual response onset latency as the time when the average firing rate exceeded the activity during control period 1 by +3 SD for 30 consecutive steps. The magnitude of visual responses was defined as the difference between the mean firing rate during 0–50 ms after visual response onset latency (peak amplitude of the visual response) and mean firing rate of control period 1 for each neuron. The delay period and perisaccadic period were defined as −100 to +40 ms around fixation offset and ±50 ms around saccade onset, respectively. To evaluate the sustained activity of LIP neurons, the activity during the delay period was compared to the activity of control period 1 using a paired $t$ test (ipsilesional lbIPS: $n = [9–64]$, contralesional lbIPS: $n = [9–32]$). To evaluate inhibitory modulation in case of TSs outside the RF, the activity during the delay period was compared with the activity of the period from 300 to 50 ms before target onset (control period 2) using a paired $t$ test (ipsilesional lbIPS: $n = [8–43]$, contralesional lbIPS: $n = [8–24]$). In this case, we set the control period to exclude the buildup activity around target onset. To detect the spatial selectivity around saccade onset, the activity of LIP neurons during the perisaccadic period for saccades toward their RF was compared with the activity during the same period for saccades toward the target presented at a point symmetrical to their RF using a $t$ test (ipsilesional lbIPS: $n = [18–107]$, contralesional lbIPS: $n = [20–56]$). In monkey C, single-unit recordings were made briefly before determining the position for muscimol injection. Four and three neurons that showed visual responses in the step saccade task were recorded in the ipsi- and contralesional lbIPS, respectively.

**Microinjection of muscimol and behavioral analysis.** Microinjection of muscimol, a GABA_A receptor agonist, was made at 3−5 sites inside the lbIPS (Supplementary Fig. 4, Supplementary Table 1) with a recording microsyringe (3-MR-S202; Crist Instrument Co., Inc.), which enabled us to record neural activity before injection. Locations of muscimol injection were determined according to the MRI images (Supplementary Fig. 4). Muscimol was dissolved in saline at a concentration of 2 or 5 μg/μL. The total amount of muscimol solution in each experiment was 1.5–2.5 μL (0.5 μL at each injection site). The speed of injection was set at 0.1 μL/25 s to avoid mechanical damage from injection pressure. The injection sites were selected from the area in which task-related activity was recorded. Data in monkey U are shown in Fig. 5. We also recorded 3–4 neurons during the step saccade task from the bilateral lbIPSs in monkeys C and U as described above, which showed similar patterns of activity (Supplementary Fig. 6). To evaluate the effects of lbIPS inactivation, we required the subject to perform the step saccade task. In most of the injection sessions, two out of three eccentricities (5, 10, and 15°) matched with the RF eccentricity of neurons recorded around the injection sites were selected for target eccentricity. In each block of trials, targets were presented at a single fixed eccentricity and at one of the eight possible locations, each of which was separated by 45° from the adjacent location. Behavioral sessions were started at 10–30 min after the injection and continued for 1–2 h.

Trials with a failure to maintain fixation on the FP and those with anticipatory saccades (saccadic reaction time <100 ms) were excluded from the following analysis. Trials in which the gaze of the monkey moved to the target within 1 s after target presentation and reached the target within 200 ms were regarded as a success, for which we checked that the monkey made a single saccade to the target later in the offline process and confirmed that double step saccades were excluded properly. The target window was a circle with a radius of half the distance between neighboring target locations (radius = eccentricity × sin [45/2]) so that there is no overlap between target windows. Success trials were defined as the trials in which saccades directly landed in the target window within 1000 ms after the fixation offset. Success rate was calculated as the percentage of the success trials to all trials. To analyze the effects of inactivation, we selected one target location with the smallest (success rate [after] − success rate [before]) value by inactivation in the affected and intact hemi-visual field separately as representative target locations. To evaluate the multimodal distribution of the endpoints of saccades in direction and eccentricity after muscimol injection, we used Silverman's test (R package: Florian Schwaiger, Hajo Holzmann). In this test, to estimate the number of modes for the distribution of data, each test was started from mode = 1 in the null hypothesis. If the number of modes = 1 was rejected by a $p$ value <0.05, the number of modes was incremented one by one, and the number of modes was determined as the first number at which the test hypothesis was not rejected. However, if a local mode was based on a single saccade, we removed the saccade as an outlier and retested multimodality. In the case judged as multimodal by the Silverman's test, the saccades in the mode which was closest to the target location were considered as "on-target" saccades and the remaining saccades were judged as "off-target". If the distribution was unimodal, all the saccades were regarded as on-target.

**Statistics and reproducibility.** As described above, spatial preprocessing and statistical analysis of the data obtained by PET imaging were performed by using FSL 5.0.9 (FMRIB Software Library, Oxford University)[55] and FEAT[53,54]. Silverman's test[56,57] for multimodality was performed using R package[58] (Florian Schwaiger, Hajo Holzmann: https://www.mathematik.uni-marburg.de/~stochastik/R_packages/silvermantest_manual.pdf; R version 3.6.0). All other analyses were performed by using MATLAB and its Statistics and Machine Learning Toolbox (Mathworks Inc., USA). All statistical tests were two-tailed unless specified otherwise and the significance level was $p < 0.05$. Details of the statistical analysis were described in each section.

**Reporting summary.** Further information on research design is available in the Nature Research Reporting Summary linked to this article.

## Data availability

Data for the figures and tables are included in Supplementary Data 1 and 2, respectively. Additional data supporting the findings of this study are available from corresponding authors upon reasonable request.

## Code availability

The codes supporting the findings of this study are available from corresponding authors upon reasonable request.

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

## Acknowledgements

This study was supported by JST-CREST to T. Isa. and H.O., and Grants-in-Aid for Scientific Research by MEXT to T. Isa. (Grant Nos. 26112008, 26221003) and H.O. (No. 26112003). We thank T. Okada and S. Urayama for taking the postmortem brain images with 7T MRI. We thank N. Takahashi, M. Kawahara, K. Takada, K. Isa, and M. Togawa for their technical assistance.

## Author contributions

T. Ikeda, M.Y., H.O., and T. Isa designed the PET experiments. T. Ikeda and R.K. conducted the PET experiments. T. Ikeda, R.K., K.O., and T.H. analyzed the PET data. R.K. and T. Isa designed the single-unit recording and reversible inactivation experiments. R.K. conducted these experiments and analyzed the data. R.K., T. Isa, and T. Ikeda mainly wrote the manuscript.

## Competing interests

The authors declare no competing interests.
