## [Peer Review File · Communications Biology]

Reviewers' comments:

Reviewer #1 (Remarks to the Author):

This study investigates activity in the brain following relatively large, unilateral lesions to the primary visual cortex in the macaque. For PET imaging, authors designed a task in which saccade frequency was varied so that they could image responsive regions of the brain to "saccade-related visuomotor activity". The PET rCBF signal showed that some areas (LIP, MT and MST) ipsilateral to the lesion became sensitive to saccade frequency only after the lesion. They then monitored the activity of a relatively small number of neurons in the lateral intraparietal area (LIP) and showed that they responded strongly to visual stimuli presented into the scotoma. Finally, they demonstrated that inactivation of area LIP on either the same or opposite side of the V1 lesion caused monkeys to act as if they were guessing where the target was when presented into the scotoma.

The study is piecemeal and data-sparse, but nonetheless innovative and of broad interest. There are shortcomings in each of the components, but together the results are fairly convincing. The existing data are sufficient to open up new ideas about how the V1-lesioned brain is able direct the eyes toward visual targets presented into the scotoma in blindsight. The similar effects of inactivating ipsilateral vs. contralateral LIP on the saccades into the scotoma is particularly interesting.

Major points

PET Imaging. The PET results are interesting but imprecise with no obvious advantage over fMRI in this application. The important PET result is that area LIP (perhaps - or LIP/MT/MST, the localization of which are somewhat uncertain) only started to significantly follow the parameter of saccade frequency following the lesion. This is interesting, though the link to blindsight is somewhat indirect.

Analysis of the main PET contrast, namely the parametric affect of changing saccade frequency in blocks, is shown in Figure 4. This figure could be improved. I think the main point is just that the slopes become significant after the lesion. To really establish that there was indeed a change, the authors should do a direct comparison of pre- vs. post-lesion regressions and test for interactions. Also, it is unclear what to make of the line offsets on the y-axis, which leads to confusion because one of the most conspicuous parts of the graph. Does the absolute value of the "Corrected rCBF (a.u.)" have any meaning? If so, what?

Their use of maximum contrast to define their ROI is also somewhat circular and wouldn't be acceptable on its own in the mainstream fMRI literature. The methodological uncertainties also doesn't do much to convince the reader that the main effect is, in fact, localized LIP.

To summarize, if the data in Figures 3 and 4 were presented alone, the results would be very unsatisfying and difficult to interpret. Luckily, there are more parts to the study.

The small number of single units is also a shame, since the results from those of cells is striking and potentially very important for understanding V1-independent visual processing.

There is very little information, or should I say proof, about the extent of the lesion. This could be done from their post-lesion anatomical scan if there is no histological data available.

Reviewer #2 (Remarks to the Author):

The study by Kato et al. addresses a central topic in visual neuroscience and investigates the neural substrates enabling visuo-motor functions following V1 damage. In fact, spared abilities to respond to visual stimuli presented in the damaged visual field have been reported in human patients (blindsight). The authors take profit of a non-human primate model of blindsight, and examine 3 macaque monkeys with surgical removal of V1 in a visuo-motor task requiring the animals to saccade toward targets either in the intact or damaged visual field. Behavioral, functional neuroimaging (PET) and neurophysiological (cell recording) methods were applied. The key finding concerns the involvement of the LIP in the damaged as well as the intact hemisphere in visuomotor task, therefore indicating the contribution of inter-hemispheric communication and the recruitment of LIP in the intact hemisphere in the type of V1-independent visuomotor functions studied here. This finding was supported by correlational evidence from PET, single unit recordings in LIP showing response to saccades executed to stimuli projected in the ipsilateral as well as contralateral targets. Finally, the causal contribution of bilateral LIP was further corroborated by temporal inactivation.

The paper is timely and interesting, multiple methods are used and results offer a reasonably convergent picture, conclusions are justified.

Overall, I appreciate the study and I believe it provides a valuable contribution to the understanding of the neural bases of V1-Independent visuo-motor functions. I have only relatively minor comments to offer, which are mainly related to pinpoint parallel and/or coherent findings previously reported in the investigation of human blindsight.

1) As summarized, the key finding concerns the contribution of bilateral LIP to visually guided saccades toward targets in the affected field. As cogently pointed out by the Authors, the result highlights an important inter-hemispheric compensatory mechanisms, whereby the intact hemisphere (particularly contralesional LIP in this case) is recruited to sustain V1-independent visuomotor functions. This inter hemispheric mechanisms has been previously documented in an fMRI study on human blindsight, which possibly represents the closest parallel to the present findings in monkeys (Celegnin et al., 2017 PNAS). In fact, that paper also investigated visuo-motor functions (not saccades but manual reaction times) and also found increased activity in the LIP in the intact hemisphere. This may be worth considering in the context of the present study and raises the possibility to perform some additional analyses on the data already collected, as suggested below

2) Is there any evidence that activity in the contralesional LIP is driven by (or is successive to) activity in the ipsilesional LIP? This would suggest indeed that the intact hemisphere is recruited to compensate for visuo-motor functions especially when the damaged hemisphere is challenged by incoming visual stimuli. Another way to test this surmise would be to compute trial-by-trial correlation in the activity of the two LIP to assess if they vary in parallel...Similarly, it could be interesting to establish possible interactions of either LIP areas with FEF. I am aware the paper is already methodologically rich, and I am not requesting to collect additional data, but to perform some analyses on the existing data set that might provide additional qualification to the reported mechanism

3) Was any anatomical measure taken (either tractography or histology if the animals have been sacrificed) that could indicate some changes in callosal fibers connecting parietal regions?

4) The authors reported that "In one session, the error saccades were directed to a location symmetrical to the target location in the affected field ..., which might be due to a deficit in visuomotor

transformation." This sounds extremely interesting, and would possibly deserve future investigation, as it seems to index a non-random effect. Perhaps, it would be relevant to note that the same behavior has been previously reported in human blindsight (Smits et al., 2019 Neuropsychologia).

5) I am not sure about the task of monkey U when the Authors say "overlap saccade task". Is it the same one reported for the other two monkeys? Please clarify, also because the single unite activity was recorded only in monkey U, so it is relevant to establish how well the different measures from PET and electrophysiology in different animals offer an integrated picture under the glue of the same task

Marco Tamietto

Reviewer #3 (Remarks to the Author):

Review of :COMMSBIO-20-1600-T , Kato...Isa, Takeda

General

The authors look at the effect of unilateral lesions of V1 on the intraparietal sulcus (IPS), studying both the ipsilesional and contralesional side, using PET O15 blood flow analysis, and single neuron analysis. They found in two monkeys ipsilesional IPS had a significantly increased bloodflow, and the contralesional LIP had a significantly increased blood flow in 1 monkey. They studied 7 ipsilesional neurons in the IPS and 9 contralesional neurons in the IPS activity in the overlap saccade task. They found that muscimol inactivation of either ipsilesional and contralesional LIP made visually guided saccades into the affected visual field less accurate.

There are a number of critical problems with the paper.

1. The contralesional blood flow increase was found in only one monkey of two. They need a tie breaker monkey.
2. It is difficult to know from their scans how much of the visual field was affected by their lesion, and whether the lesions overlapped V2. They could have mapped the striate and prestriate cortex using functional MRI before the lesions, and then mapped the lesions on their visual field map. Now we don't know the degree of V1 ablation
3. They say that they recorded from LIP, but the criteria they used are very weak: "Recording sites were estimated based on the relative positions between the lateral bank of the intraparietal sulcus and the recording chamber on magnetic resonance images" It is difficult from their drawings to know that they were not in MIP and PRR. It would be better if they showed us some clearer evidence – MnCl₂ injections from the muscimol syringe visible in an MRI, for example. Now the best accurate localization is to the intraparietal sulcus (IPS) not to LIP.
4. The visual latencies of their putative LIP neurons, even on the contralesional side, were longer than many studies have reported, and the spike density functions do not show the usual presaccadic increase shown in LIP. It is unlikely that these were from the real LIP, although they could have come from MIP or PRR in the intraparietal sulcus.
5. The authors are correct that their IPS lesions have little effect on visually guided and memory guided saccades – but LIP inactivation causes a profound neglect when monkeys are rewarded for making either ipsilateral or contralateral saccades to two simultaneously present presented targets. They inevitably choose the ipsilateral target unless there is a large time delay between the appearance of the ipsilateral and contralateral saccade (Wardak).
6. The eye movements after muscimol injections are woefully underanalyzed. In the first place the striate lesions themselves seem to have caused some saccadic inaccuracy. This is not mentioned – instead the authors say "However, they regained their ability to localize a target in the affected field

by saccades within a few months." In the second place, although the bimodal distribution data are convincing, we don't know if the even the on-target mode is made worse by IPS inactivation.

7. They use a metric of saccadic accuracy called "success rate." Unfortunately I could not find a definition of success rate in the paper. They have to define the metric.

In sum, the result, that intraparietal sulcus inactivation, both ipsi- and contralesional, to a striate lesion affect saccade accuracy is an interesting finding. Note that unlike the authors, I did not use the word 'LIP' because of the non-LIPness of their neuronal recordings and the inadequacy of their anatomical location. However the paper needs to be revised significantly as discussed above.

Reviewers' comments:

We thank all the reviewers for sparing your precious time to review our manuscript. The major changes added to the original manuscripts are as follows;

1. We added more detailed data on the extent of V1 lesion in Figure 1 and SFig.1.
2. In response to the comments from Reviewer#3, we decided not to call the sites for electrophysiological recording and pharmacological inactivation "LIP" but instead renamed as "lateral bank of the intraparietal sulcus (lbIPS)" and discussed that at least it partly overlaps the LIP. Accordingly, we also changed the title from "lateral intraparietal sulcus" to "posterior parietal cortex" to be more general.
3. We added more detailed analysis on the PET data, including task dependent connectivity (SFig.9) to study functional connectivity of bilateral IPS.
4. We added more detailed analysis on the saccadic behavior; analyzed the accuracy of saccades after the V1 lesion and before the lbIPS inactivation (SFig.8). We also analyzed the on-target saccades after the lbIPS inactivation (STable 2 and 3).

Below, we described our point-by-point responses to individual comments from the reviewers.

Reviewer #1 (Remarks to the Author):

This study investigates activity in the brain following relatively large, unilateral lesions to the primary visual cortex in the macaque. For PET imaging, authors designed a task in which saccade frequency was varied so that they could image responsive regions of the brain to "saccade-related visuomotor activity". The PET rCBF signal showed that some areas (LIP, MT and MST) ipsilateral to the lesion became sensitive to saccade frequency only after the lesion. They then monitored the activity of a relatively small number of neurons in the lateral intraparietal area (LIP) and showed that they responded strongly to visual stimuli presented into the scotoma. Finally, they demonstrated that inactivation of area LIP on either the same or opposite side of the V1 lesion caused monkeys to act as if they were guessing where the target was when presented into the scotoma.

The study is piecemeal and data-sparse, but nonetheless innovative and of broad interest. There are shortcomings in each of the components, but together the results are fairly convincing. The existing data are sufficient to open up new ideas about how the V1-lesioned brain is able direct the eyes toward visual targets presented into the scotoma in blindsight. The similar effects of inactivating ipsilateral vs. contralateral LIP on the saccades into the scotoma is particularly interesting.

(Our responses)

Thank you for sparing time for reviewing our article. Yes, the data are relatively sparse because of many limitations to the experiments, but we believe the combination of the present experimental results would open up new ideas about the mechanism of blindsight.

Major points

PET Imaging. The PET results are interesting but imprecise with no obvious advantage over fMRI in this application. The important PET result is that area LIP (perhaps - or LIP/MT/MST, the localization of which are somewhat uncertain) only started to significantly follow the parameter of saccade frequency following the lesion. This is interesting, though the link to blindsight is somewhat indirect.

Analysis of the main PET contrast, namely the parametric affect of changing saccade frequency in blocks, is shown in Figure 4. This figure could be improved. I think the main point is just that the slopes become significant after the lesion. To really establish that there was indeed a change, the authors should do a direct comparison of pre- vs. post-lesion regressions and test for interactions. Also, it is unclear what to make of the line offsets on the y-axis, which leads to confusion because one of the most conspicuous parts of the graph. Does the absolute value of the "Corrected rCBF (a.u.)" have any meaning? If so, what?

Their use of maximum contrast to define their ROI is also somewhat circular and wouldn't be acceptable on its own in the mainstream fMRI literature. The methodological uncertainties also doesn't do much to convince the reader that the main effect is, in fact, localized LIP.

(Our responses)

We agree with the reviewer's opinion that a direct comparison of pre- vs. post-lesion regression is important. We revised Figs.3 and 4 to demonstrate this point more clearly. Figs.4a and b now show the result of whole brain analysis for an interaction between task and lesion (pre vs post), where significant increase of rCBF was found in the MCC/SMA and bilateral IPS areas both in axial (Fig.4a) and coronal sections (Fig.4b). Figure 4c and d show plots of the rCBF response across task variables in pre- and post-lesion periods at selected voxels in a part of the significant areas in Fig.4a-b. The main points of these analyses are to find significant areas in the interaction analysis first, and to see whether the task dependency in those areas had actually increased in the post-lesion period. As the reviewer pointed out, previous Fig.4 was somewhat circular, therefore, we removed any statistics from these plots in Fig. 4c-d. However, the results clearly show the positive relationships between task condition and rCBF in ipsi- and contralesional IPS areas in post-lesion period.

Y-axis in the Fig.4c and d showed rCBF values after regressing out the effect of global signal. However, it is difficult to compare the value across different conditions since 1) absolute value of "corrected rCBF" is not measurable in this PET system, 2) rCBF may be highly biased regionally by partial volume mainly due to low resolution of PET scanner. Thus, the offsets on the y-axis are not comparable across subjects, ROIs, or lesion periods. However, note that it is valid to analyze the slopes between task conditions within-subjects AND within-period sessions.

To summarize, if the data in Figures 3 and 4 were presented alone, the results would be very unsatisfying and difficult to interpret. Luckily, there are more parts to the study.

The small number of single units is also a shame, since the results from those of cells is striking and potentially very important for understanding V1-independent visual processing.

(Our response)

Let us clarify why we recorded only a small number of neurons. This is because the main purpose of the recordings was to determine the injection sites of muscimol. For this purpose, the number of recordings were minimal to avoid damage to the brain tissue before the successive microinjection experiments for inactivation. In spite of this shortcomings, we decided to report these data because we thought that the data were quite valuable.

There is very little information, or should I say proof, about the extent of the lesion. This could be done from their post-lesion anatomical scan if there is no histological data available.

(Our response)

In accordance with the reviewer's comment, we added the data showing the extent of the lesion in three monkeys in Fig.1 as additional panels and Supplementary information as SFig.1. See detail for SFig.1. Briefly, we showed histological section of the brain of monkey T and MR images of formalin-fixed brain in monkeys C and U. Based on these data, we made line drawings indicating the lesion site in the sections which include the calcarine sulcus and added to Fig.1d, f, and h. In these drawings, we compared the ipsilesional and contralesional sides and delineated the lesion extent and indicated on the sections for the contralesional (intact) side. By adding these data, it is now evident that the caudal surface of V1 and the caudal bank of the calcarine sulcus was completely removed. Thus, we conclude that the V1 areas representing 2 deg – 25 deg eccentricities were removed.

Reviewer #2 (Remarks to the Author):

The study by Kato et al. addresses a central topic in visual neuroscience and investigates the neural substrates enabling visuo-motor functions following V1 damage. In fact, spared abilities to respond to visual stimuli presented in the damaged visual field have been reported in human patients (blindsight). The authors take profit of a non-human primate model of blindsight, and examine 3 macaque monkeys with surgical removal of V1 in a visuo-motor task requiring the animals to saccade toward targets either in the intact or damaged visual field. Behavioral, functional neuroimaging (PET) and neurophysiological (cell recording) methods were applied. The key finding concerns the involvement of the LIP in the damaged as well as the intact hemisphere in visuomotor task, therefore indicating the contribution of inter-hemispheric communication and the recruitment of LIP in the intact hemisphere in the type of V1-independent visuomotor functions studied here. This finding was supported by correlational evidence from PET, single unit recordings in LIP showing response to saccades executed to stimuli projected in the ipsilateral as well as contralateral targets. Finally, the causal contribution of bilateral LIP was further corroborated by temporal inactivation.

The paper is timely and interesting, multiple methods are used and results offer a reasonably convergent picture, conclusions are justified.

Overall, I appreciate the study and I believe it provides a valuable contribution to the understanding of the neural bases of V1-Independent visuo-motor functions. I have only relatively minor comments to offer, which are mainly related to pinpoint parallel and/or coherent findings previously reported in the investigation of human blindsight.

(Our response)

Thank you for all the helpful comments. We carefully included the responses to the comments in our revised manuscript.

1) As summarized, the key finding concerns the contribution of bilateral LIP to visually guided saccades toward targets in the affected field. As cogently pointed out by the Authors, the result highlights an important inter-hemispheric compensatory mechanisms, whereby the intact hemisphere (particularly contralesional LIP in this case) is recruited to sustain V1-independent visuomotor functions. This inter hemispheric mechanisms has been previously documented in an fMRI study on human blindsight, which possibly represents the closest parallel to the present findings in monkeys (Celeghin et al., 2017 PNAS). In fact, that paper also investigated visuo-motor functions (not saccades but manual reaction times) and also found increased activity in the LIP in the intact hemisphere. This may be worth considering in the context of the present study and raises the possibility to perform some additional analyses on the data already collected, as suggested below

(Our response)

In response to the suggestion, we added the argument on the interhemispheric interaction in the blindsight subjects by referring to Celeghin et al. (2017).

“Another possibility is that the contralesional lIPs contributed to visuomotor processing for saccade execution through inter-hemispheric interactions as suggested for human blindsight subject⁴³.” (page 15, line 396-398)

2) Is there any evidence that activity in the contralesional LIP is driven by (or is successive to) activity in the ipsilesional LIP? This would suggest indeed that the intact hemisphere is recruited to compensate for visuo-motor functions especially when the damaged hemisphere is challenged by incoming visual stimuli. Another way to test this surmise would be to compute trial-by-trial correlation in the activity of the two LIP to assess if they

vary in parallel...Similarly, it could be interesting to establish possible interactions of either LIP areas with FEF. I am aware the paper is already methodologically rich, and I am not requesting to collect additional data, but to perform some analyses on the existing data set that might provide additional qualification to the reported mechanism

(Our response)

This is an important question, and we think the interhemispheric interaction should be tested further. Therefore, we tested interactions between rCBF of interest (at ipsi- or contralesional IPS), task and lesion, which is often called as a psychophysiological interactions (PPIs) (Friston et al., 1997). Using either of IPS areas as a seed region, we analyzed whether its functional connectivity is altered by task AND lesion. Although we did not find functional connectivity between bilateral IPS, several regions including midcingulate cortex (MCC) and contralesional midbrain showed significant PPIs in both seeds of ipsi- and contralesional IPS, suggesting the former regions might be involved in the residual visuomotor function by mediating interhemispheric interaction between bilateral IPS. We add this result to the Results and Discussion, with an additional supplementary figure (SFig.9).

“In addition, to search regions which showed altered task- and lesion-dependent connectivity with IPS areas, we analyzed psychophysiological interactions (PPI)^{27, 28}, using ipsi- and contralesional IPS seeds (same ROIs as in Fig.4d-e). The results showed significant increase in PPI in the post-lesion period compared to the pre-lesion period in MCC, contralesional midbrain, retrosplenial cortex, and contralesional primary motor cortex, with both ipsi- and contralesional IPS areas (SFig.9).” (page 12, line 307-312)

“Indeed, our PPI analysis showed changes in task-dependent connectivity between MCC and bilateral IPS areas in post-lesion period (SFig.9), suggesting that MCC-IPS interactions mediated the residual visuomotor function.” (page 15, line 419-422)

3) *Was any anatomical measure taken (either tractography or histology if the animals have been sacrificed) that could indicate some changes in callosal fibers connecting parietal regions?*

(Our response)

Anatomical changes in callosal fibers are very interesting. We added the data of the extent of the lesion in three monkeys in response to the comment from Reviewer #1. As described in our response to the Reviewer #1's comments, we had the parasagittal sections of the caudal half of the brain of monkey T, which were processed with Nissl staining. For monkeys C and U, the brains fixed with formalin remained and we processed these brains with 7T MRI in our facility. The images were good enough to convincingly delineate the lesion extent. However, Nissl-stained sections of monkey T were not prepared for evaluation of thickness of callosal fibers connecting bilateral LIPs. Concerning the MR data, we are still on the way to develop the techniques for reliable diffusion tensor tractography on the postmortem macaque brain. Furthermore, to obtain quantitative analysis, a certain number of intact brains for comparison are needed. Thus, although the suggestions are quite precious for us, we think such analysis should stand as a separate project for the future. (Fig.1 and SFig.1)

4) *The authors reported that “In one session, the error saccades were directed to a location symmetrical to the target location in the affected field ..., which might be due to a deficit in visuomotor transformation.” This sounds extremely interesting, and would possibly deserve future investigation, as it seems to index a non-random effect. Perhaps, it would be relevant to note that the same behavior has been previously reported in human blindsight (Smits et al., 2019 Neuropsychologia).*

(Our response)

Thank you very much. We added the description on this interesting symptom together with

reference to Smits et al. (2019).

“In one session, the error saccades were directed to a location symmetrical to the target in the affected field (SFig.5, ContU2), which might be related to the previous report in human blindsight²³ and may be due to deficit in visuomotor transformation.” (page 10, line 267-270)

5) *I am not sure about the task of monkey U when the Authors say “overlap saccade task”. Is it the same one reported for the other two monkeys? Please clarify, also because the single unite activity was recorded only in monkey U, so it is relevant to establish how well the different measures from PET and electrophysiology in different animals offer an integrated picture under the glue of the same task.*

(Our response)

In the PET study and inactivation experiments, we used the round saccade or step visually guided saccade task (with no delay between the fixation offset and target presentation) to clarify the areas contributing to the performance of visually guided saccades. But here, it is not clear whether the neurons in the areas carry the visual information or saccade motor activity. To address this question, the overlap saccade task, in which the delay is inserted between the target presentation and fixation offset, was used to dissociate the visual and motor response components in the recorded cells. The target area in the monkey U was adjusted to the same part of the lIPs in which the PET study showed the association between the rCBF and the task conditions. We added the description regarding this issue.

“We sampled task-related neurons during an overlap saccade task to dissociate the visual and motor responses in ipsi- and contralesional lIPs in monkey U (SFig.4, blue rectangles).” (page 7, line 171-172)

Marco Taitto

Reviewer #3 (Remarks to the Author):

Review of :COMMSBIO-20-1600-T, Kato...Isa, Takeda

General

The authors look at the effect of unilateral lesions of V1 on the intraparietal sulcus (IPS), studying both the ipsilesional and contralesional side, using PET O15 blood flow analysis, and single neuron analysis. They found in two monkeys ipsilesional IPS had a significantly increased bloodflow, and the contralesional LIP had a significantly increased blood flow in 1 monkey. They studied 7 ipsilesional neurons in the IPS and 9 contralesional neurons in the IPS activity in the overlap saccade task. They found that muscimol inactivation of either ipsilesional and contralesional LIP made visually guided saccades into the affected visual field less accurate.

There are a number of critical problems with the paper.

1. *The contralesional blood flow increase was found in only one monkey of two. They need a tie breaker monkey.*

(Our response)

It is not practical to add a tie breaker monkey as it will require a few more years for PET experiment. However, we could find consistent results in two monkeys by focusing on the group analysis to avoid false negative. Detailed voxel-wise analysis for each subject showed very similar results in contralesional IPS: monkey C showed significant task relation in post-lesion period ($z = 3.51$) and also significant task-lesion interaction ($z = 3.84$) in slightly posterior IPS ($x = 10.5$, $y = -4$, $z = 28$), while monkey T showed significant task relation in post-lesion period ($z = 2.33$) and also significant task-lesion interaction ($z = 2.41$) in slightly anterior IPS ($x = 8.5$, $y = -1.5$, $z = 26$). Slight misalignment might be due to the

structural difference between two monkeys. We revised Figure 4 to emphasize the task-lesion interaction in bilateral IPS areas. Therefore, we considered that bilateral IPS areas are involved in visuomotor function more in the post lesion period, and this hypothesis was supported by the following inactivation experiments.

2. It is difficult to know from their scans how much of the visual field was affected by their lesion, and whether the lesions overlapped V2. They could have mapped the striate and prestriate cortex using functional MRI before the lesions, and then mapped the lesions on their visual field map. Now we don't know the degree of V1 ablation

(Our response)

As described in our response to Reviewer #2, in accordance with the reviewer's comment, we added the data showing the extent of the lesion in three monkeys in Fig.1 as additional panels and Supplementary information as SFig.1. See detail for SFig.1. Briefly, we showed histological section of the brain of monkey T and MR images of formalin-fixed brain in monkeys C and U. Based on these data, we made line drawings indicating the lesion site in the sections which include the calcarine sulcus and added to Fig.1d, f, and h. In these drawings, we compared the ipsilesional and contralesional sides and delineated the lesion extent and indicated on the sections for the contralesional (intact) side. By adding these data, it is now evident that the caudal surface of V1 and the caudal bank of the calcarine sulcus was completely removed. Thus, we conclude that the V1 areas representing 2 deg – 25 deg eccentricities were removed.

3. They say that they recorded from LIP, but the criteria they used are very weak: "Recording sites were estimated based on the relative positions between the lateral bank of the intraparietal sulcus and the recording chamber on magnetic resonance images" It is difficult from their drawings to know that they were not in MIP and PRR. It would be better if they showed us some clearer evidence – MnCl2 injections from the muscimol syringe visible in an MRI, for example. Now the best accurate localization is to the intraparietal sulcus (IPS) not to LIP.

4. The visual latencies of their putative LIP neurons, even on the contralesional side, were longer than many studies have reported, and the spike density functions do not show the usual presaccadic increase shown in LIP. It is unlikely that these were from the real LIP, although they could have come from MIP or PRR in the intraparietal sulcus.

(Our response to the Comments 3 and 4)

At the time of these experiments, we did not use $MnCl_2$ images to localize the injection sites (Now it is our standard to use gadolinium). We modified the SFigure 4 to show the recording/injection sites overlaying on the MRI images of the brain and grid for fixing the injection needles. The grid system does not allow the lateral shift of the needles. Therefore, we are sure that we recorded and injected in the lateral bank of the IPS (lbIPS in the text).

As pointed out by the reviewer, the presaccadic increase in activity was not clear in our data shown in Figure 5, which would not satisfy the criterion of the LIP neurons presumably because the delay time of the overlap saccade task was too short. Instead, we added examples of the recorded cells which suggest the existence of presaccadic activity in the step saccade task (SFig.6). These results, together with the MRI images in the new SFig.4, suggest that the current recording/injection sites were in the lateral bank of IPS and at least partly overlap the LIP. However, it is true that we have to be careful with precise use of names of brain areas. Therefore, we decided to operationally rename the sites as "the lateral bank of the intraparietal sulcus (lbIPS)" all through the manuscript, figures and tables. We also changed the words of the title from "lateral intraparietal area" to more generally "posterior parietal cortex".

5. The authors are correct that their IPS lesions have little effect on visually guided and memory guided saccades – but LIP inactivation causes a profound neglect when monkeys

are rewarded for making either ipsilateral or contralateral saccades to two simultaneously present targets. They inevitably choose the ipsilateral target unless there is a large time delay between the appearance of the ipsilateral and contralateral saccade (Wardak).

(Our response)

Thank you very much for pointing this out. The main point of our current study was to show the effect of IbIPS inactivation on the simple task. This is an interesting issue on the function of the posterior parietal cortex. Wardak et al. showed little effect on visually guided and memory guided saccades but profound effect on detection of the contralateral target in the double-target presentation task during IbIPS inactivation. We added the related description to the Discussion section.

“In intact monkeys, the LIP is more involved in the target detection/selection from a complex background rather than a simple reflexive saccade as shown in neural activity^{24,39,40} and reversible inactivation experiment²². After V1 lesion, visual information becomes more ambiguous and less reliable, which makes a simple saccade task more complex and difficult like search task with distractors and a double-target task in the intact animals, which might require the LIP.” (page 14, line 372-377)

We think it is possible that V1 lesioned monkeys showed neglect-like effects by IbIPS inactivation in our task as their vision is less reliable in their contralateral visual field during IbIPS inactivation. However, our task was not designed to study the neglect in particular and that should be studied in the future.

6. The eye movements after muscimol injections are woefully underanalyzed. In the first place the striate lesions themselves seem to have caused some saccadic inaccuracy. This is not mentioned – instead the authors say “However, they regained their ability to localize a target in the affected field by saccades within a few months.” In the second place, although the bimodal distribution data are convincing, we don’t know if the even the on-target mode is made worse by IPS inactivation.

(Our response)

Thank you very much for pointing this out.

As clearly described in the behavioral data during the PET experiments (SFig.2) and our previous article (Yoshida et al. 2008), the V1 lesion itself makes saccades inaccurate even long after the lesion. We added the analysis of the saccades during the control sessions. Since the data were obtained long after the lesion and not exactly in the same task design and in the different experimental setups (the behavioral data before the lesion was obtained in the PET facility and then the monkeys were moved to different institutions where the recording/inactivation experiments were conducted), direct comparison with the data before the lesion was not possible. Therefore, we compared the distribution of saccade endpoints in the intact and affected hemifield by Steel-Dwass test and showed that the saccades were less accurate in the affected visual field (SFig.8). Furthermore, the change in accuracy of the “on-target” saccades were analyzed for both direction and eccentricity of saccades and shown in STable 2 and 3. Here, even if the data of the separated mode (“off-target saccades”) were excluded, the direction and the eccentricity distribution of saccade endpoints of “on-target” saccades became less accurate after muscimol injections.

Thus, although V1 lesion itself caused saccade inaccuracy, the effect of IbIPS inactivation was clarified by the large reductions of success rate and the increased variance of the saccade endpoint distribution (the multimodal distribution and increased IQR) (Fig.6 and 7, SFig.7, STable1, 2 and 3). The results were described in page 10, line 260 – 276 of the main text and also in STables 2 and 3 and their explanation in the Supplementary materials.

7. They use a metric of saccadic accuracy called “success rate.” Unfortunately I could not find a definition of success rate in the paper. They have to define the metric.

(Our response)

Thank you very much for pointing this out. We added descriptions about how to calculate the success rate in Method section (page 25, line 698-701).

“Success trials were defined as the trials in which saccades directly landed in the target window within 1000 ms after the fixation offset. Success rate was calculated as the percentage of the success trials to all trials.” (page 24, line 672-674)

In sum, the result, that intraparietal sulcus inactivation, both ipsi- and contralesional, to a striate lesion affect saccade accuracy is an interesting finding. Note that unlike the authors, I did not use the word ‘LIP’ because of the non-LIPness of their neuronal recordings and the inadequacy of their anatomical location. However the paper needs to be revised significantly as discussed above.

REVIEWERS' COMMENTS:

Reviewer #1 (Remarks to the Author):

The authors have addressed my comments adequately.

My main opinion hasn't change much, that this manuscript is useful and publishable in its present form, despite each of its individual components having shortcomings that make interpretation difficult. The main new finding is that a portion on the lateral bank of the intraparietal sulcus, probably LIP, receives visual input from the V1 scotoma in the same hemisphere and also contributes critically to visually-guided saccade behavior. I think the preponderance of evidence makes that case.

The modifications in the revision are relatively minor. I am happy that the authors now present more information about the extent of the lesion. That said, the additions to Figure 1 are not spectacular, nor is the presentation of the histology and postmortem in Figure S1. Nonetheless, its presence is an improvement.

Minor point:

Figure 1 legend, does not reflect new panels (e.g. "(b: monkey T, c: monkey C, and d: monkey 870 U)").

Reviewer #2 (Remarks to the Author):

The authors did a thorough job in addressing my previous comments. Although the new results are not always consistent with the predictions, the overall picture offers an integrated multi-method view of the issues that is both fair and comprehensive. I do not have further comments to offer and I think the paper stands in good shape for publication
Marco Tamietto

Reviewer #3 (Remarks to the Author):

I think that the authors have answered most of my concerns. I wish that in analyzing saccade accuracy they had used a direct analysis of endpoint variability as well as mode. This would make the paper stronger, but I would leave this to the authors's discretion.

Responses to REVIEWERS' COMMENTS:

Reviewer #1 (Remarks to the Author):

The authors have addressed my comments adequately.

My main opinion hasn't change much, that this manuscript is useful and publishable in its present form, despite each of its individual components having shortcomings that make interpretation difficult. The main new finding is that a portion on the lateral bank of the intraparietal sulcus, probably LIP, receives visual input from the V1 scotoma in the same hemisphere and also contributes critically to visually-guided saccade behavior. I think the preponderance of evidence makes that case.

The modifications in the revision are relatively minor. I am happy that the authors now present more information about the extent of the lesion. That said, the additions to Figure 1 are not spectacular, nor is the presentation of the histology and postmortem in Figure S1. Nonetheless, its presence is an improvement.

Minor point:

Figure 1 legend, does not reflect new panels (e.g. "(b: monkey T, c: monkey C, and d: monkey 870 U)").

(Our response)

Thank you for pointing this out. We were careless about this issue. Now we corrected the Figure 1 legend as follows;

“Figure 1. Lesion extent in the individual animals.

a. The area of V1 in horizontal planes (colored in red). **b.** Horizontal levels of the section in each panel of **a** with the corresponding numbers, overlaid on a sagittal brain slice. **c., e.** and **g.** The extent of the V1 lesion in monkeys T, C and U is indicated on the horizontal planes in black, respectively. The horizontal levels of the individual horizontal sections in **c, e** and **g** are matched with those in **a.** **d., f.** and **h.** The sagittal planes across the calcarine sulcus (cal) on the contralesional (left panel) and ipsilesional (right panel) side in monkey T, C and U, respectively. The extent of V1 is indicated in red. Posterior (leftward) side of the blue lines (indicated in the arrow) in the left panels was considered to be lesioned on

the ipsilesional side (right panel). Abbreviations: cal, calcarine sulcus; cc, central sulcus; ip, intraparietal sulcus; lu, lunate sulcus; st, superior temporal sulcus.”

Reviewer #2 (Remarks to the Author):

The authors did a thorough job in addressing my previous comments. Although the new results are not always consistent with the predictions, the overall picture offers an integrated multi-method view of the issues that is both fair and comprehensive. I do not have further comments to offer and I think the paper stands in good shape for publication

Marco Tamietto

(Our response)

Thank you for your insightful comments and sparing your precious time for our manuscript.

Reviewer #3 (Remarks to the Author):

I think that the authors have answered most of my concerns. I wish that in analyzing saccade accuracy they had used a direct analysis of endpoint variability as well as mode. This would make the paper stronger, but I would leave this to the authors's discretion.

(Our response)

We actually directly analyzed the endpoint variability (see Supplementary Table 2 and 3).

Thank you for your insightful comments and sparing your precious time for our manuscript.